# Impairment of lipid homeostasis causes lysosomal accumulation of endogenous protein aggregates through ESCRT disruption

**John Yong, Jacqueline E Villalta, Ngoc Vu, Matthew A Kukurugya†, Niclas Olsson, Magdalena Preciado López, Julia R Lazzari-Dean, Kayley Hake, Fiona E McAllister, Bryson D Bennett, Calvin H Jan***

Calico Life Sciences LLC, South San Francisco, United States

**\*For correspondence:**
cjan@calicolabs.com

**Present address:** †Department of Molecular & Cell Biology, University of California, Berkeley, United States

## eLife Assessment

Protein and lipid homeostasis is essential for maintaining cellular functions but their crosstalk remains largely unknown. This **important** manuscript deals with this interesting topic and applies the powerful unbiased tools of somatic cell genetics to discover evidence suggesting a link between sphingolipids/cholesterol ester metabolism and lysosomal protein aggregation. The authors provide **compelling** orthogonal evidence to support their conclusions.

**Abstract** Protein aggregation increases during aging and is a pathological hallmark of many age-related diseases. Protein homeostasis (proteostasis) depends on a core network of factors directly influencing protein production, folding, trafficking, and degradation. Cellular proteostasis also depends on the overall composition of the proteome and numerous environmental variables. Modulating this cellular proteostasis state can influence the stability of multiple endogenous proteins, yet the factors contributing to this state remain incompletely characterized. Here, we performed genome-wide CRISPRi screens to elucidate the modulators of proteostasis state in mammalian cells, using a fluorescent dye to monitor endogenous protein aggregation. These screens identified known components of the proteostasis network and uncovered a novel link between protein and lipid homeostasis. Increasing lipid uptake and/or disrupting lipid metabolism promotes the accumulation of sphingomyelins and cholesterol esters and drives the formation of detergent-insoluble protein aggregates at the lysosome. Proteome profiling of lysosomes revealed ESCRT accumulation, suggesting disruption of ESCRT disassembly, lysosomal membrane repair, and microautophagy. Lipid dysregulation leads to lysosomal membrane permeabilization but does not otherwise impact fundamental aspects of lysosomal and proteasomal functions. Together, these results demonstrate that lipid dysregulation disrupts ESCRT function and impairs proteostasis.

## Introduction

Protein aggregation is a hallmark of aging and age-related pathologies, including multiple neurodegenerative diseases and amyloidoses that affect individual or multiple organs (*Chiti and Dobson, 2017*; *Eisenberg and Jucker, 2012*). Dozens of proteins that are normally soluble may form aggregates that are traditionally associated with specific diseases – e.g. tau tangles and amyloid plaques with Alzheimer's Disease, and transthyretin deposits with transthyretin amyloidosis (*Chiti and Dobson, 2017*). Although specific pathogenic proteins are often studied in isolation, a more complex

relationship among different aggregation-prone proteins exists, either directly or through interaction with the cellular machinery regulating proteostasis. For instance, experiments in yeast, worms, and mammalian cell culture have demonstrated considerable cross-talk and cross-sensitization between classic aggregation-prone proteins (*Gidalevitz et al., 2009*; *Guo et al., 2013*). Further, a diverse set of proteins exhibit reduced solubility and lose their native structure during normal aging (*David et al., 2010*). Intriguingly, these insoluble age-associated proteins can seed the formation of classical amyloids in vitro, and contribute to functional decline in aging animals (*Huang et al., 2019*). Together, the evidence suggests that systemic changes in the state of cellular proteostasis may underlie proteinopathy and highlights the importance of expanding the scope of studies beyond specific disease-associated protein species.

The core proteostasis network is well described and involves the synthesis, folding, trafficking, and degradation of proteins (*Hipp et al., 2019*; *Kaushik and Cuervo, 2015*; *Labbadia and Morimoto, 2015*). Most of these components closely follow the life cycle of proteins, including (1) pathways that modulate protein synthesis and degradation, such as mTORC, ribosome biogenesis, the ubiquitin proteasome system, and autophagy; (2) molecular chaperones, which aid protein folding; and (3) stress response pathways that regulate the activity of (1) and (2) in response to accumulation of unfolded or misfolded proteins, such as ER-associated degradation and the unfolded protein response. Nevertheless, the components and network topology vary depending on cell type and cell state. Furthermore, mechanisms that underlie shifts in cellular proteostasis state, especially ones that do not directly involve core nodes of the proteostasis network, remain largely unknown.

Unbiased, whole-genome screening is a powerful tool for discovering cellular components and pathways that are involved in modifying a specific phenotype. Previous studies have applied genetic screens in different cell models to uncover, for specific proteins, modifiers of their aggregation, toxicity, or propensity for aggregation to propagate from seeds. These systems are often driven by overexpressing aggregation-prone proteins and/or seeding with extracellular pre-formed fibers (*Kramer et al., 2018*; *Chen et al., 2019*; *Willingham et al., 2003*). Thus, while much progress has been made in understanding the homeostasis of several discrete proteins, the mechanisms influencing the broader cellular proteostasis network remain largely unknown. In particular, we are interested in cellular mechanisms that affect the aggregation state of endogenously expressed proteins, as these may be the most germane to age-associated loss of proteostasis.

To uncover novel components and pathways that regulate endogenous protein homeostasis, we applied CRISPR interference (CRISPRi) technology to perform unbiased, genome-wide screens on K562 cells using an aggresome-staining dye. ProteoStat is known to label both experimentally induced aggresomes (*Shen et al., 2011*) and aggregates that accumulate with age in the CNS of mammals (*Leeman et al., 2018*; *Gefen et al., 2015*; *Vonk et al., 2020*). To our knowledge, ours are the first CRISPRi genome-wide screens performed on unmodified mammalian cells to interrogate endogenous protein aggregation. These screens revealed a link between lysosomal lipid homeostasis and proteostasis. Increased levels of sphingomyelins and cholesterol esters, either as a result of increased lipid uptake or decreased lysosomal degradative capacity, lead to increased protein aggregation predominantly colocalized with the lysosomes. Proteomic analysis of isolated lysosomes revealed accumulation of detergent-insoluble ESCRT (endosomal sorting complex required for transport) components, indicating potential impairment of lysosomal membrane repair and microautophagy. Lipid dysregulation induces or potentiates lysosomal membrane permeabilization (LMP), the rescue of which partially restores proteostasis. Other aspects of lysosomal and proteasomal functions essential for proteostasis (lysosomal pH, lysosomal protease activity, and proteasome activity) were unaffected by lipid dysregulation. Together, this work provides strong evidence that supports an emergent role of lipid dysregulation in driving protein aggregation through disruption of the ESCRT pathway.

## Results

### Measurement of endogenous proteostasis state by flow cytometry

ProteoStat is used to stain endogenous protein aggregates in fixed cells. Like Thioflavin T, ProteoStat fluorescence increases when rotationally confined by binding to cross-beta sheets in amyloidal structures. Recent studies have used the dye to visualize aggregated proteins in mouse NSCs and *C. elegans* (*Leeman et al., 2018*; *Shen et al., 2011*; *Vonk et al., 2020*; *Bohnert and Kenyon, 2017*). To

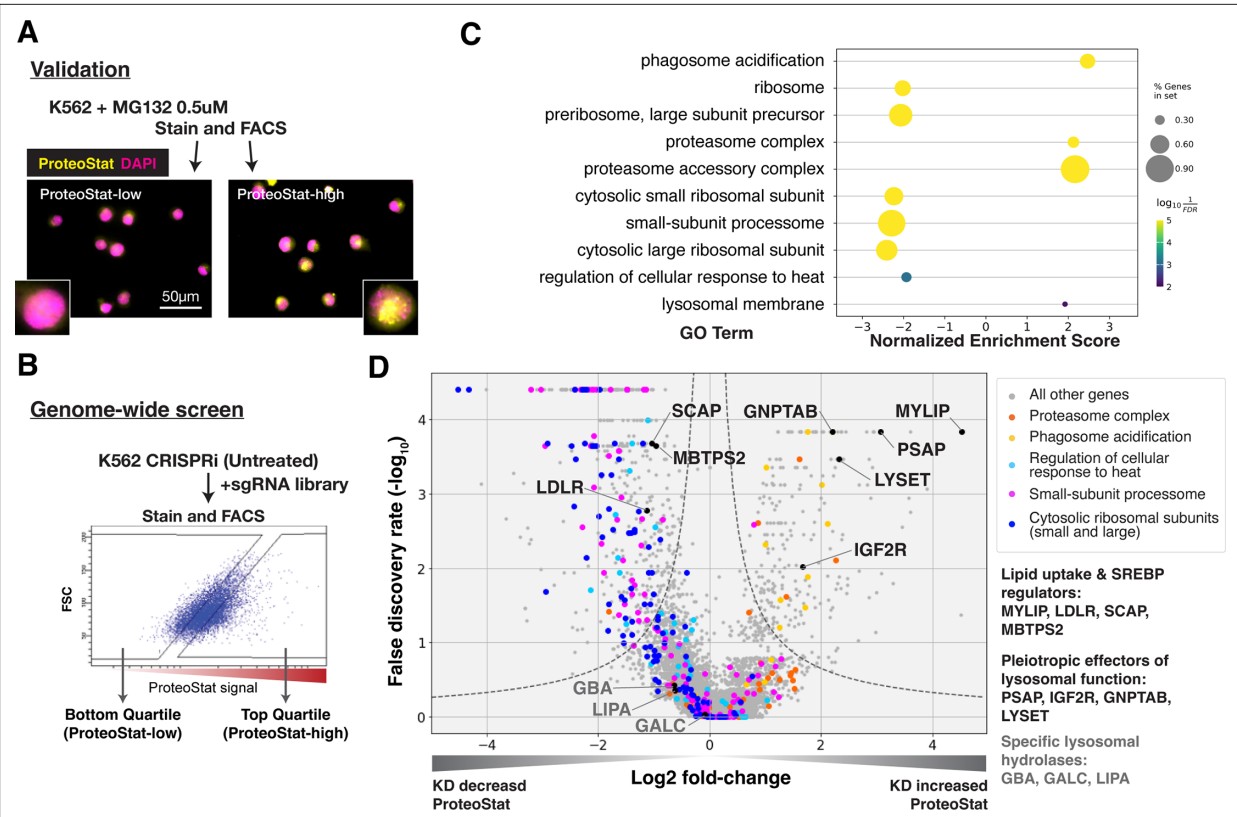

**Figure 1.** CRISPRi screens identified major cellular proteostasis components and implicated lipid uptake and metabolism pathways. (**A**) ProteoStat validation: MG132-treated cells were stained, sorted into top and bottom quartiles (ProteoStat-high and ProteoStat-low, respectively), and imaged. (**B**) CRISPRi screen schematic: Untreated K562 cells were infected with sgRNA library, selected, stained, and sorted using gates that corrected for correlation between staining intensity and cell size. (**C**) A subset of GO Terms (Biological Process and Cellular Component) identified with GSEA on screen results. Positive Enrichment scores indicate an increased likelihood of genes in the GO Term to increase ProteoStat when knocked down. (**D**) Volcano plot highlighting major proteostasis pathways and select genes involved in lipid uptake and metabolism. Dotted line denotes where the product of |$\log_2$(fold-change)| and -$\log_{10}$(False discovery rate) is constant and equals to 1*-$\log_{10}$(0.05).

The online version of this article includes the following source data and figure supplement(s) for figure 1:

**Source data 1.** CRISPRi ProteoStat screen MAGeCK analysis combining inputs from two biological replicates; gene-level output.

**Figure supplement 1.** ProteoStat screen validation: reagent, reduced-scale batch retest, and chaperone test.

**Figure supplement 1—source data 1.** CRISPRi ProteoStat batch retest sgRNA counts depicted in *Figure 1—figure supplement 1B*.

verify that ProteoStat can report changes in proteostasis in human cell culture, we inhibited proteasomes in K562 cells and stained them for microscopy or flow cytometry analysis. As expected, proteasome inhibition by MG132 increased ProteoStat intensity, with numerous bright puncta around the nucleus (*Figure 1—figure supplement 1A*). ProteoStat-high cells obtained by fluorescence-activated cell sorting (FACS) from cells treated with a lower dose of MG132 – which generated mixed phenotypes – likewise showed numerous bright puncta, whereas ProteoStat-low cells displayed diffuse staining of lower intensity, indicating that this tool can be used to report and isolate cells in different proteostasis states (*Figure 1A*).

## Genome-wide screen for modifiers of ProteoStat recovered major known components of the proteostasis machinery

We performed FACS-based, pooled screens in K562 CRISPRi cells, which stably expressed KRAB-dCas9. Briefly, cells carrying individual sgRNAs were fixed, stained, and sorted by ProteoStat intensity into top and bottom quartiles (ProteoStat-high and ProteoStat-low, respectively; *Figure 1B*). Illumina sequencing was used to quantify the frequency of each sgRNA in each of these two populations. We

performed this experiment twice and evaluated the effect of each gene's knockdown (KD) on proteostasis using the MAGeCK package (*Li et al., 2014*).

KDs of 239 genes increased ProteoStat, whereas KDs of 607 genes decreased it (out of 20,704 protein-coding genes targeted by the sgRNA library; FDR <0.05). We validated the results of a subset of genes by repeating the screen at a reduced scale (batch retest; *Figure 1—figure supplement 1B*). Gene set enrichment analysis (GSEA) captured ontologies in which genes showed consistent effects in the same direction, even though individual genes may fall below an artificial threshold. Many of these genes and pathways are known proteostasis modifiers (*Figure 1C–D*). These include ribosomal proteins (both small 40 S and large 60 S subunits), ribosome biogenesis and processing, proteasome subunits, lysosomal proton pump subunits (vATPases), and certain chaperones. KD of many ribosomal proteins and ribosome assembly factors decreased ProteoStat, consistent with reduced input of proteins into the proteostasis network (*Stein and Frydman, 2019*). Knocking down subunits of the proteasome complex increased ProteoStat, consistent with MG132 treatment in our validation experiments. Disrupting vATPases similarly increased ProteoStat, likely via deacidification and impairment of lysosomes, an important center where multiple catabolic processes converge. These effects on ProteoStat following inhibition of proteasome and lysosomal acidification were replicated using pharmacological inhibitors (*Figure 1—figure supplement 1C*). In contrast, KD of many chaperones, including HSPA4, HSP90AB1, DNAJB6, and several subunits of TRiC chaperonins, paradoxically reduced ProteoStat. Perturbing chaperones can trigger the newly described safeguard against protein aggregation mechanism that targets certain newly synthesized aggregation-prone proteins to the lysosome, notably before large visible aggregates are formed (*Jung et al., 2020*). Alternatively, initial HSP70 or HSP90 inhibition can upregulate heat shock response as a negative feedback (*Kijima et al., 2018*; *Akerfelt et al., 2010*; *Ali et al., 1998*). Indeed, pharmacological inhibition of chaperones increased ProteoStat after 30 min, but decreased after 24 hr (*Figure 1—figure supplement 1C*).

Surprisingly, KD of most of the major components directly involved in macroautophagy did not affect ProteoStat in either direction. These include the core autophagosome formation components (e.g. ATG5, ATG10, ATG12, and the ATG8 protein family) and the autophagy adaptors (NBR1, TAX1BP1, OPTN, SQSTM1 etc). Although we performed our screens under nutrient-replete conditions, perturbations of these macroautophagy genes in K562 under similar growth conditions have previously been demonstrated to influence autophagic flux (*Shoemaker et al., 2019*). These results suggest that basal macroautophagy plays a negligible role in clearing endogenous amyloid-like structures in this system.

## Lipid uptake and metabolism modulate proteostasis

The strong association of lysosomal vATPases with Proteostat phenotype is likely due to the need to maintain a lysosomal pH of ~4.5–5.0 for optimal activity of the hydrolases that break down proteins, lipids, and nucleic acids. Failure to maintain this acidic pH can cause lysosomal storage of undegraded macromolecules. Accordingly, in addition to the vATPases, several genes that broadly regulate lysosomal function impacted ProteoStat. These include genes that modulate the delivery or activity of multiple lysosomal enzymes (e.g. GNPTAB, IGF2R, LYSET, and PSAP) or lipid uptake and metabolism (e.g. LDLR, MYLIP, and SCAP; *Figure 1D*). However, no perturbations of more discrete activities (e.g. individual cathepsins, lipid hydrolases – e.g. GBA, LIPA, and GALC – or solute carriers) did so, suggesting that proteostasis is generally resilient and becomes impaired only in response to pleiotropic perturbations.

KD of low-density lipoprotein receptor (LDLR) reduced ProteoStat. Correspondingly, KD of the E3 ubiquitin-protein ligase MYLIP, which targets LDLR for sterol-regulated degradation, led to one of the strongest signal increases. MYLIP KD increased ProteoStat via upregulating LDLR levels rather than through other MYLIP targets, as the effect of LDLR KD was epistatic to MYLIP in cells with double KD (*Figure 2A*). Cultured cells take up LDL – lipoproteins rich in cholesterol esters – through LDLR and the endo-lysosomal pathway. Once inside the acidic environment of lysosomes, cholesterol ester is hydrolyzed into free fatty acid and cholesterol, the latter of which is trafficked to the ER. Excess cholesterol exerts negative feedback on LDLR levels via two mechanisms: (1) oxysterols (oxidized derivatives of cholesterol) activate the liver X receptor (LXR), in turn increasing MYLIP expression and LDLR degradation (*Zelcer et al., 2009*); (2) high ER cholesterol sequesters the SCAP-SREBPs complex, suppressing its activity and in turn reducing LDLR expression as well as de novo cholesterol synthesis

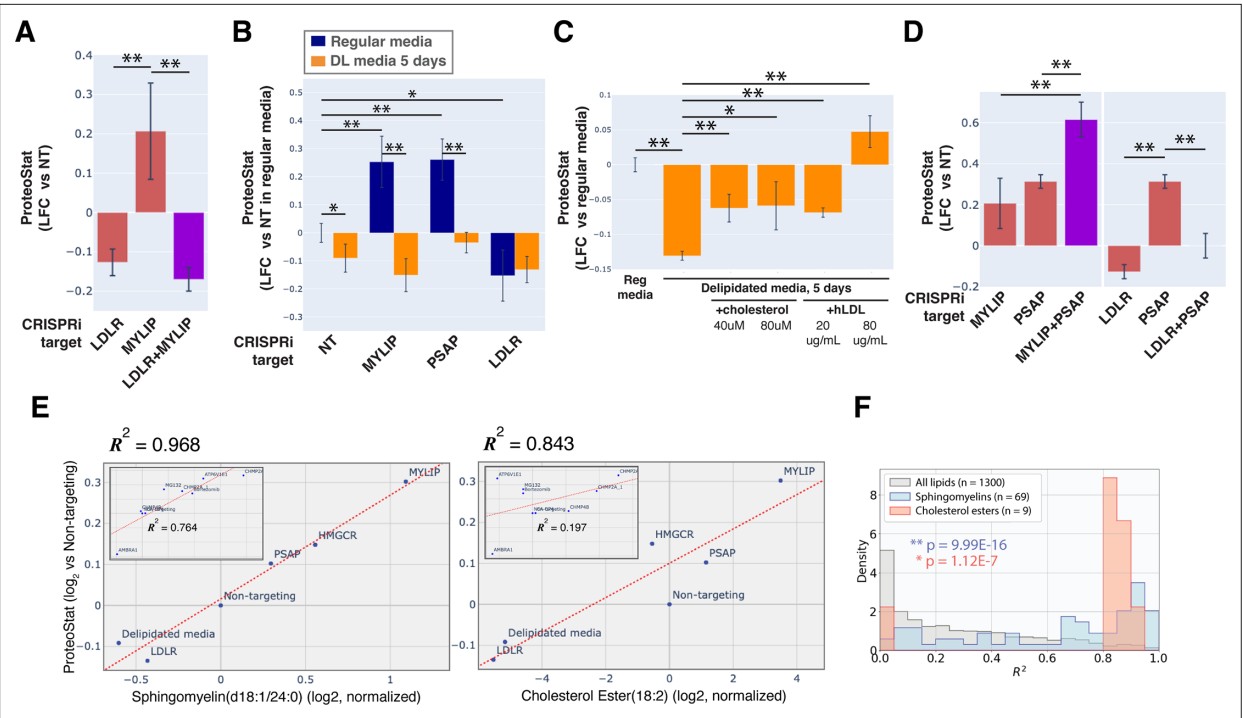

**Figure 2.** Accumulation of sphingolipid and cholesterol ester can impair proteostasis. (**A & D**) Comparing effects on ProteoStat of single vs double KDs (n≥2) (**B**) Effects of gene KDs on ProteoStat in cells maintained in media with regular or delipidated FBS (DL media). (n=3) (**C**) Effects of supplementing free cholesterol or human LDL in restoring ProteoStat, which was lowered in cells grown in DL media (n=3). Phenotypes in A-D were normalized first to an internal staining control, and then to cells carrying non-targeting guides. (✳: $p<0.05$; ✳✳: $p<0.01$; t-test between pairs of samples as indicated with horizontal lines). (**E**) ProteoStat vs abundance of indicated lipid species in cells with KD of a core set of lipid-related target genes, or including other ProteoStat perturbing genes (**insets**). Both ProteoStat and lipid abundance are averages of biological triplicates and normalized to cells carrying non-targeting guides. Least squares linear regression models (dotted red lines) were generated and the Coefficient of determination, $R^2$, values are shown. (**F**) Histograms of $R^2$ values for all lipid species within each indicated lipid class as in E. p-values were calculated using K-S test comparing $R^2$ values within each lipid class versus that of all lipids. The number of individual lipid species analyzed is indicated for each lipid class.

The online version of this article includes the following source data and figure supplement(s) for figure 2:

**Source data 1.** Lipidomics results and ProteoStat phenotype depicted in *Figure 2E–F*.

**Figure supplement 1.** ProteoStat does not directly stain cholesterol nor major components of the plasma membrane.

**Figure supplement 2.** Example of lipid classes whose levels do not correlate with ProteoStat, and investigating the effects of perturbing individual saposins on ProteoStat.

(*Goldstein et al., 2006*). Two positive regulators of the SREBP pathways (SCAP and MBTPS2) also reduced ProteoStat when knocked down (*Figure 1D*). In contrast, genes along the mevalonate – de novo cholesterol synthesis – pathway did not consistently score in our screen, indicating the difference between cholesterol that is taken up vs synthesized de novo.

Chronic depletion of exogenous LDL using media that contains delipidated serum reduced ProteoStat to a level similar to LDLR KD and was sufficient to abrogate the effect of MYLIP KD (*Figure 2B*), further demonstrating that the effect of these genetic perturbations was mediated through lipid uptake and not direct degradative action of MYLIP on putative aggregation-prone substrates. Supplementing either exogenous cholesterol or human LDL reversed the effect of LDL depletion (*Figure 2C*). We also examined the possibility that ProteoStat may directly stain cholesterol by acutely extracting cholesterol using methyl-β-cyclodextrin (MBCD). MBCD removed most free cholesterol as determined by Filipin III staining, but did not affect ProteoStat (*Figure 2—figure supplement 1A–B*). In neither untreated nor MG132-treated cells did ProteoStat and Filipin III staining overlap, indicating that ProteoStat does not stain free cholesterol nor any major components of the plasma membrane (*Figure 2—figure supplement 1C*). Furthermore, in an in vitro biphasic lipid bilayer system (giant unilamellar vesicles, GUVs), ProteoStat stained the lipid bilayer poorly and was not brighter at the cholesterol-rich phase (*Figure 2—figure supplement 1D*). Together, the evidence suggests

that increased flux of LDL uptake and/or the levels of lipid metabolites generated during the process increases protein aggregation.

KD of genes involved in lysosomal sphingolipid metabolism and lysosomal protein trafficking (GNPTAB, IGF2R, LYSET, and PSAP) increased ProteoStat in our screens (*Figure 1D*). GNPTAB encodes two subunits of a Golgi-resident enzyme, GlcNAc-1-phosphotransferase, which catalyzes the first step to form mannose 6-phosphate (M6P) markers on lysosomal enzymes and is stabilized by the recently characterized LYSET (also named TMEM251; *Richards et al., 2022*). IGF2R (also known as the cation-independent M6P receptor) binds to these lysosomal enzymes and facilitates their transport from golgi to lysosome. PSAP, or prosaposin, is the precursor to saposin peptides, which are required to activate lysosomal sphingolipid catabolizing enzymes. The effect of PSAP KD was dramatically exacerbated by MYLIP KD and alleviated by LDLR KD (*Figure 2D*), indicating that lipid uptake can impinge on proteostasis through its interaction with lipid catabolism in lysosomes. The individual lysosomal hydrolases activated by saposins, such as GBA, GALC, and LIPA, did not rise above the detection threshold in our screen, again suggesting that the proteostasis network, as measured by ProteoStat, is generally resilient to perturbations of discrete functional nodes or changes in individual lipid species.

## Sphingolipids and cholesterol esters mediate the effect of lipid perturbations on proteostasis

Having demonstrated that lipid metabolism could influence proteostasis, we wondered if specific lipid species might be involved. To address this question, we performed lipidomic analysis on cells in which lipid metabolism was perturbed, either genetically by CRISPRi, or environmentally by lipid depletion. These perturbations were selected from the screen results to represent a range of effects on ProteoStat staining. When considering perturbations that directly affect lipid uptake and metabolism, we found remarkable correlation between ProteoStat staining and individual levels of most lipids in the sphingomyelin (SM) and cholesterol ester (CE) classes ($R^2$ of up to 0.968 and 0.843, for SM and CE, respectively, *Figure 2E*). Indeed, lipids belonging to these two classes were better correlated with the ProteoStat phenotype when compared with the full collection of lipid compounds assayed (*Figure 2F*, see *Figure 2—figure supplement 2A* for the lipid classes ceramides [Cers], lactosylceramides [LacCers], or hexosylceramides [HexCers] as counterexamples). In contrast, KD of targets that do not impinge directly on lipid metabolism (e.g. AMBRA1 or CHMP2A) or else expected to have strong pleiotropic effects beyond lipid metabolism (e.g. ATP6V1E1) did not induce strong correlations between SMs/CEs and ProteoStat (insets in *Figure 2E*). These results further highlight the multimodality of proteostasis regulation, where lipid dysregulation represents one, among other, major components.

To determine if any lipid species alone contributed critically to protein aggregation induced by lipid dysregulation, we leveraged the modularity of the highly conserved PSAP. Upon delivery to the lysosome, the proprotein is cleaved proteolytically into four active peptides, saposins, (Saps) A, B, C, and D. Saps are involved in the hydrolysis of glycosphingolipids (GSLs) as enzymatic activators, facilitating the interaction between soluble lysosomal hydrolases and GSLs embedded in the membrane (*Kolter and Sandhoff, 2005*; *Xu et al., 2010*). Distinct residues within the Sap-hydrolase interface confer hydrolase specificity on each saposin (e.g. SapA binds to Galactocerebrosidase, GALC, and stimulates galactosylceramide hydrolysis, *Figure 2—figure supplement 2B*; *Hill et al., 2018*). Deficiency in full-length PSAP or individual Saps (by truncation, inactivating mutations, or reduced expression) lead to various lysosomal storage disorders, with neuropathology and often dysfunction of other organs (*Schulze and Sandhoff, 2011*). Depending on which Sap is deficient, distinct GSL species are found to accumulate in the storage cells. We therefore performed rescue experiments using different PSAP variants to perturb turnover of individual GSLs more specifically. We expressed full length PSAP or variants with either a single Sap or all-but-one Sap (*Figure 2—figure supplement 2C*). Full-length PSAP largely rescued the increase in ProteoStat induced by PSAP KD (p=0.07 vs non-targeting, no-rescue control), while the variant with all four Saps inactivated (PSAP ΔABCD) failed to rescue (p=0.74 vs PSAP KD, no-rescue control; *Figure 2—figure supplement 2D*). Furthermore, addition of any single Sap did not fully rescue the KD phenotype. Accordingly, inactivating any single Sap also did not fully abrogate PSAP's ability to rescue. Together, the evidence suggests that proteostasis impairment from sphingolipid accumulation is pleiotropic and is likely attributable to the action of multiple but not a single lipid. This is consistent with the observation

that, in our CRISPRi screens, none of the individual GSL hydrolases significantly increased Proteo-Stat level.

## Lipid dysregulation leads to accumulation of insoluble lysosomal protein aggregates

Lipid-related perturbations that impair proteostasis suggested a central role for lysosomes, prompting us to further characterize ProteoStat staining, especially its subcellular localization. Aggresomes are reportedly composed of misfolded and aggregated proteins that are poly-ubiquitinated and transported along microtubules to coalesce into perinuclear assemblies. These aggresomes can be recognized by autophagy receptors to undergo canonical macroautophagy and lysosomal degradation (*Kocaturk and Gozuacik, 2018*). Alternatively, cytosolic or organelle cargo may be recruited and directly delivered into endolysosomal compartments in a process termed microautophagy that is dependent on the ESCRT pathway (*Kuchitsu et al., 2023*; *Oku et al., 2017*; *Schäfer et al., 2020*). ProteoStat was initially reported to colocalize with mono-/poly-ubiquitinated proteins, p62, and LC3, and has been used as a general reporter for protein aggregation (*Bohnert and Kenyon, 2017*; *Leeman et al., 2018*; *Shen et al., 2011*; *Vonk et al., 2020*). Additionally, lysosome-accumulating ProteoStat-positive aggregates were reported in quiescent neural stem cells but not their activated counterparts (*Leeman et al., 2018*), nor in other rapidly dividing cell types.

We initially characterized aggresomes induced by pharmacological proteasome inhibition. Treatment with MG132 for 4 hr induced ProteoStat puncta that colocalized predominantly with the lysosomes, in addition to a diffuse but weaker signal in the cytosol (*Figure 3—figure supplement 1A–B*). A similar distribution was observed after 20 hr of treatment. In contrast, mono-/poly-ubiquitinated proteins increased only modestly at 4 hr; at 20 hr these signals increased globally and did not concentrate in the lysosomes. Punctate structures of mono-/poly-ubiquitinated proteins became more apparent at a higher dose of MG132. These puncta were non-lysosomal and ProteoStat-positive, though their ProteoStat intensities were lower than the lysosomal ProteoStat puncta (*Figure 3—figure supplement 1A*, white arrowheads). In comparison, proteasome inhibition by Bortezomib led to proportionally stronger cytoplasmic ProteoStat and mono-/poly-ubiquitination and less pronounced lysosomal puncta. Finally, when lysosomes were deacidified with Bafilomycin A1, we observed strong overlapping signals of ProteoStat and mono/poly-ubiquitination that were partially colocalized with the lysosomes, consistent with impaired ability to remove and degrade ubiquitin chains or ubiquitinated proteins. Patterns of Thioflavin T staining were similar to ProteoStat (*Figure 3—figure supplement 1C–D*). These results with diverse inhibitors demonstrate subtle pharmacological biases – with MG132 favoring lysosomal puncta and Bortezomib causing both classical aggresomes and lysosomal phenotypes – and the ability for ProteoStat to capture these variations. These observations of lysosomal ProteoStat-high ubiquitination-low puncta, together with the aforementioned lack of evidence of macroautophagy involvement, are consistent with the microautophagy pathway where cargos are processed by deubiquitinating enzymes (STAMBP and UBPY) associated with ESCRT during transport into lysosomes (*Hanson et al., 2009*). Under mild proteostatic stress, ubiquitinated proteins accumulate and are transported into lysosomes by microautophagy, while being deubiquitinated in the process. When proteostatic stress prolongs or intensifies, deubiquitinating capacity may be overwhelmed, resulting in increased accumulation of ubiquitinated proteins both inside and outside lysosomes. Indeed, UBPY KD increased lysosomal signals of both ProteoStat and mono-/poly-ubiquitinated proteins, consistent with a role for microautophagy in delivery of misfolded proteins to the lysosome (*Figure 3—figure supplement 1E–F*).

Cells with lipid impairment (MYLIP or PSAP KD) exhibited lysosomal ProteoStat puncta that were non-ubiquitinated, in addition to occasional cytoplasmic double-positive puncta (*Figure 3A–B*, white arrowheads). This suggests a moderate level of proteostatic stress that does not overwhelm the capacity to deubiquitinate. Accordingly, cells with MYLIP and PSAP KD did not exhibit growth defects (*Figure 3—figure supplement 2A*). To verify the presence of protein aggregates in lysosomes under the above conditions, we isolated lysosomes using Lyso-IP (Lysosome Immunoprecipitation) (*Abu-Remaileh et al., 2017*). HA epitopes are targeted to the lysosomal surface by TMEM192, which enable lysosomal isolation, followed by differential extraction into RIPA (detergent-soluble) and Urea/SDS (detergent-insoluble) fractions. This orthogonal approach confirmed that lipid-perturbation (MYLIP and PSAP KDs) resulted in accumulation of detergent-insoluble protein aggregates in the lysosome

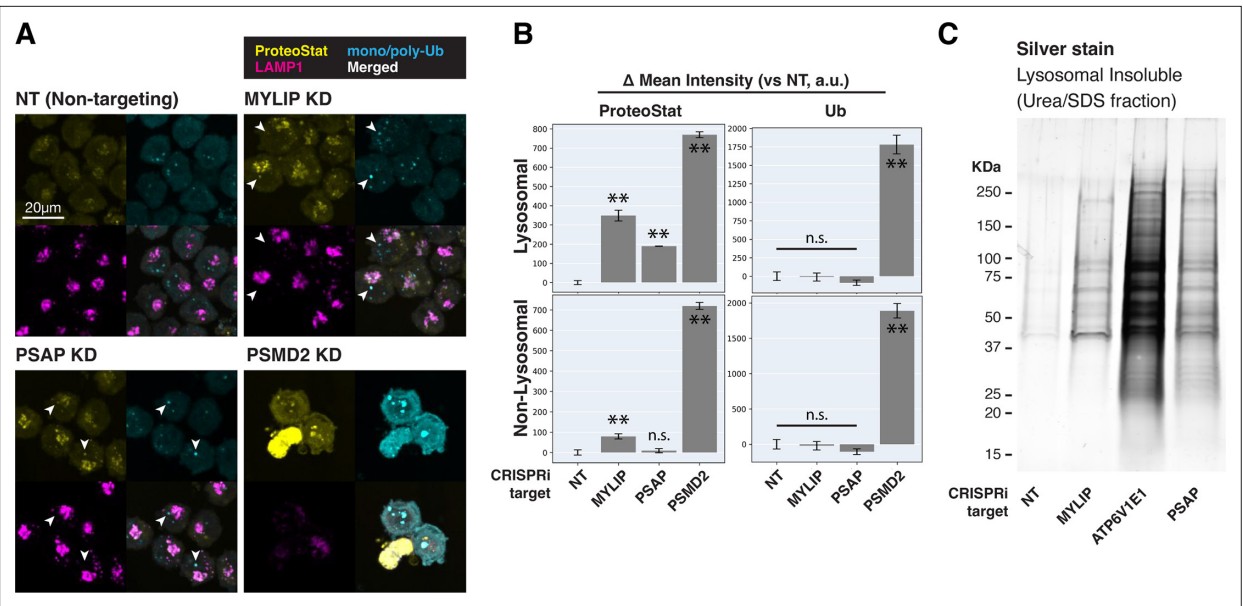

**Figure 3.** Lipid dysregulation leads to accumulation of insoluble lysosomal protein aggregates. (**A**) K562 cells with the indicated KDs were co-stained with ProteoStat and antibodies against ubiquitinated proteins or lysosomes and imaged with confocal (maximum projection of multiple-z-slices). (**B**) Quantification (n=3) of ProteoStat and mono/poly-ubiquitinated proteins (representative images shown in A) segregated according to co-localization with lysosome (LAMP1 staining). (**C**) Representative (from triplicate experiments) polyacrylamide gel with Silver Stain showing Lysosomal Insoluble fractions from cells with various KDs after Lyso-IP and differential extraction. (✳✳: p<0.01; n.s.: not significant; t-test vs NT control or between pairs of samples as indicated with horizontal lines).

The online version of this article includes the following source data and figure supplement(s) for figure 3:

**Source data 1.** PDF file containing original gel image for *Figure 3C*, indicating the treatment for each lane.

**Source data 2.** Original gel image for *Figure 3C*.

**Figure supplement 1.** Characterization of sub-cellular distribution of ProteoStat, Thioflavin T, and ubiquitinated proteins under various perturbations.

**Figure supplement 2.** Evaluating the effect of lipid perturbations on growth rate, and lysosomal protein aggregates characterization under proteasome inhibition.

**Figure supplement 2—source data 1.** PDF file containing original gel image for *Figure 3—figure supplement 2B*, indicating the treatment for each lane.

**Figure supplement 2—source data 2.** Original gel image for *Figure 3—figure supplement 2B*.

(*Figure 3C*). Inhibiting lysosomal acidification (ATP6V1E1 KD) or proteasome activity (by MG132) led to larger amounts of lysosomal aggregates (*Figure 3C*, *Figure 3—figure supplement 2B*).

## Lipid dysregulation leads to the accumulation of insoluble ESCRT components at the lysosome

We performed in-depth quantitative proteomics coupled with isobaric tandem mass tags (TMT) on different Lyso-IP fractions to determine the identity of lysosomal detergent-insoluble proteins. 67 of 70 core lysosomal resident (*Singh et al., 2020*) proteins were captured in the Lysosomal Total fraction (before differential extraction), validating the Lyso-IP-proteomics method. Within the Lysosomal Insoluble fraction, 35 proteins were over-represented (q-value <0.05) in MYLIP KD cells versus control. Remarkably, 9 of the top 12 insoluble proteins induced by MYLIP KD were components involved in the ESCRT pathway (*Figure 4A*), including multiple ESCRT-III subunits (e.g. CHMP2A, CHMP4A & B), accessory proteins (ALIX and PTPN23), and ESCRT-III disassembling enzymes (VPS4A and VTA1). The ESCRT machinery is involved in multiple cellular processes that involve membrane scission with a negative curvature (membrane bending out and away from the cytoplasm), including cytokinesis, ciliogenesis, multivesicular body formation, microautophagy, and lysosomal membrane repair. Accordingly, GSEA of insoluble lysosomal proteins enriched in MYLIP KD compared to control highlighted the Multivesicular Body gene set (GO:0005771; *Figure 4B*). This enrichment of ESCRT components was not simply a consequence of increased abundance in the Lysosomal Total fraction, as multiple

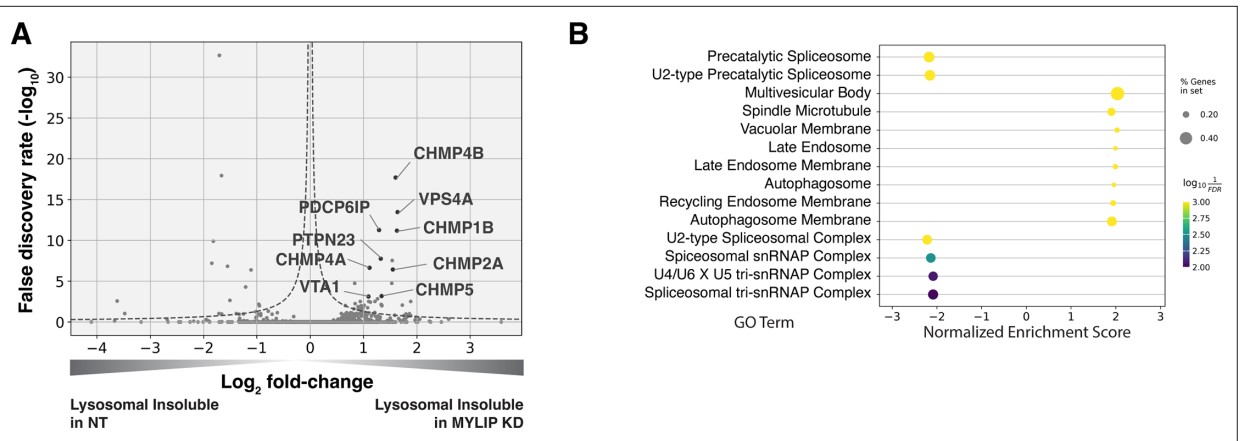

**Figure 4.** Lysosomal protein aggregates under lipid accumulation are enriched in ESCRT components. (**A**) Volcano plot of Lyso-IP proteomics experiment highlighting ESCRT components and associated proteins that became more insoluble in the lysosome in MYLIP KD vs control cells. Dotted line denotes where the product of |log₂(fold-change)| and -log₁₀(False discovery rate) is constant and equals to 1*-log₁₀(0.05). (**B**) GO Terms (Cellular Component) identified with GSEA on Lyso-IP Proteomics results. Positive Enrichment scores indicate an increased likelihood of genes in the GO Term to be enriched in Lysosomal Insoluble fraction in MYLIP KD vs control cells.

The online version of this article includes the following source data and figure supplement(s) for figure 4:

**Source data 1.** Lyso-IP proteomics results in the Lysosomal Insoluble fractions, comparing MYLIP KD vs NT control.

**Figure supplement 1.** Increased lysosomal abundance alone does not cause insolubility.

**Figure supplement 1—source data 1.** Lyso-IP proteomics results in the Lysosomal Total fractions, comparing MYLIP KD vs NT control.

proteins (most notably vATPases) were also more abundant in the total, but not insoluble, fraction of MYLIP KD cells (*Figure 4—figure supplement 1A*). ESCRT-III subunits have previously been shown to form detergent-insoluble hetero-polymers on membranes (*Hanson et al., 2008*; *McCullough et al., 2015*). These ESCRT-III polymers can be trapped on membranes by depleting VPS4A/B (*Cashikar et al., 2014*), which has the dual function of progressing ESCRT-III polymerization through subunit exchange and ultimately disassembling the complex upon completion of membrane scission (*Pfitzner et al., 2020*). Here, our results provide novel evidence that excessive SM and CE levels may disrupt ESCRT function by impeding ESCRT-III progression and/or resolution.

## Lysosomal membrane permeabilization (LMP) contributes to proteostasis impairment

Can ESCRT disruption mediate the effect of lipid dysregulation on proteostasis impairment? As ESCRT contributes to lysosomal membrane repair and ESCRT deficiency is sufficient to drive LMP, we examined the effect of lipid-perturbing KDs on lysosomal membrane integrity using a TagBFP-fused Galectin-3 (GAL3) reporter. GAL3 is normally diffused in the cytoplasm but becomes punctate upon binding to glycans on the lumenal side of the lysosomal membrane that are exposed upon damage. We imaged the reporter under a combination of candidate gene KDs and/or acute LMP induced by LLOMe (L-leucyl-L-leucine, a lysosomotropic membranolytic agent; *Uchimoto et al., 1999*). MYLIP KD and, to a much weaker degree, PSAP KD, both increased LMP, as did KDs of ESCRT-III subunits (CHMP2A and CHMP6) and ATP6V1E1 (*Figure 5A–B*). Chemically inducing LMP with LLOMe was sufficient to cause a moderate lysosomal accumulation of protein aggregates (*Figure 5C*). However, the magnitude of LMP was not directly related to ProteoStat intensity across conditions (*Figure 5C–E*), suggesting that while LMP is sufficient to promote lysosomal protein aggregation, other factors are likely to contribute.

We further tested if LMP was necessary for lysosomal protein aggregation by promoting lysosomal membrane repair through VPS4A overexpression. Chemically induced LMP was almost completely rescued by ectopic VPS4A expression (*Figure 5D*), presumably due to accelerated progression and resolution of ESCRT-III polymers. This rescue of chemically induced LMP was largely negated in the background of MYLIP KD (*Figure 5D*), suggesting that lipid dysregulation impairs ESCRT pathway in an orthogonal manner or upstream to ESCRT-III resolution. Accordingly, VPS4A overexpression only

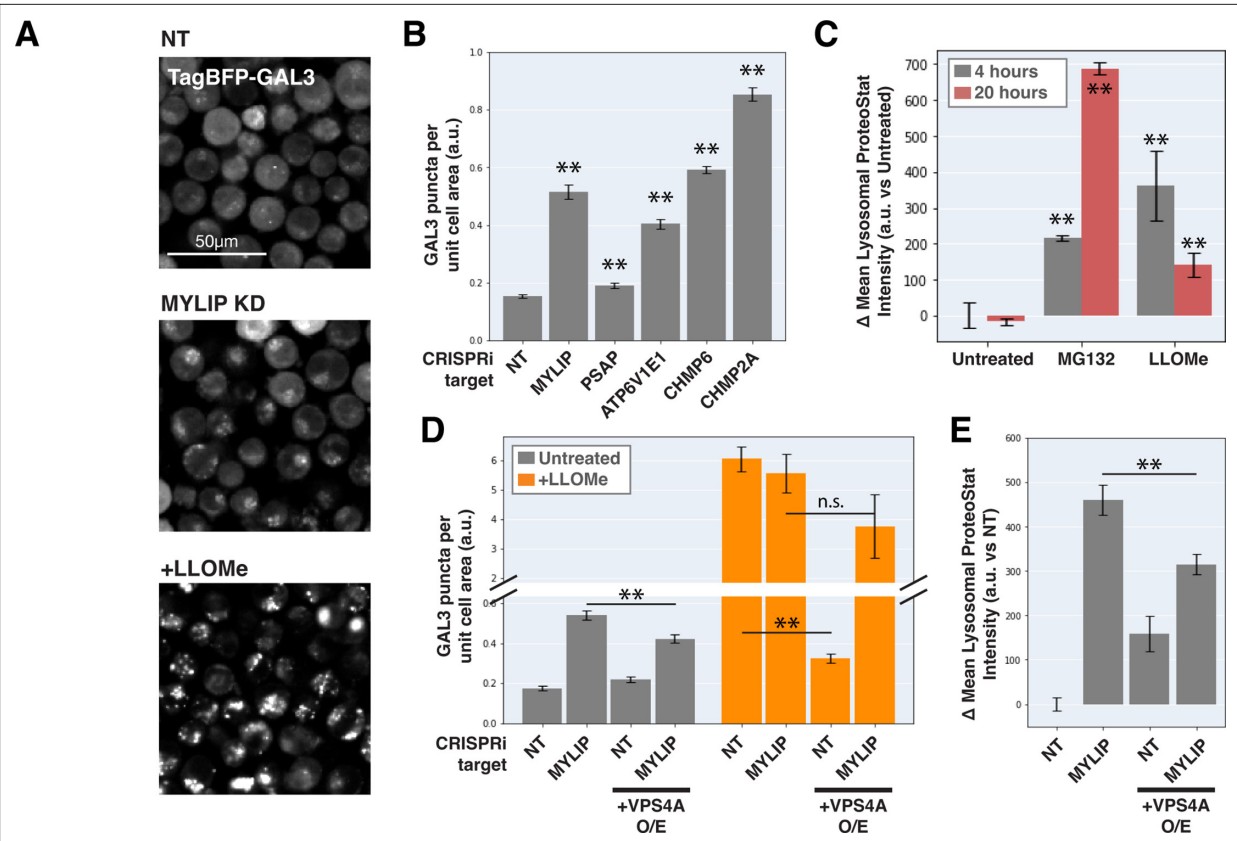

**Figure 5.** Lysosomal membrane permeabilization contributes to proteostasis impairment. (**A**) Representative fluorescence microscopy images of K562 cells expressing TagBFP-GAL3, with NT control, sgRNA targeting MYLIP, or treated with LLOMe at 120 uM for 4 hr. (**B**) Quantification (n=3) of TagBFP-GAL3 puncta in cells with various KDs. (**C**) Quantification (n=3) of lysosomal ProteoStat signal under various treatments. (**D**) Quantification (n=3) of TagBFP-GAL3 puncta in cells with a combination of control vs MYLIP KD,±LLOMe treatment (120 µM for 4 hr), and ±VPS4 A overexpression. (**E**) Quantification (n=3) of lysosomal ProteoStat signal in cells with a combination of control vs MYLIP KD and ±VPS4A overexpression. (✴: p<0.05; ✴✴: p<0.01; n.s.: not significant; t-test vs NT / Untreated control or between pairs of samples as indicated with horizontal lines).

led to a modest reduction in LMP and lysosomal protein aggregation that were induced by MYLIP KD (*Figure 5D–E*). Together, these data show that LMP contributes, albeit only partially, to lipid-induced lysosomal protein aggregation and that lipid accumulation may impair VPS4A-mediated resolution of ESCRT assemblies.

## Proteostasis impairments due to lipid dysregulation are not correlated with changes in lysosomal and proteasomal functions

We sought to determine whether lipid dysregulation impacted proteasomal or lysosomal function. SM and CE metabolisms were previously linked with several diseases with proteinopathy, including Alzheimer's disease (AD) and the lysosomal storage disease Niemann Pick's type C (NPC) (*Kirkegaard et al., 2010*; *Schulze and Sandhoff, 2011*; *van der Kant et al., 2019*; *Yamazaki et al., 2001*). Furthermore, multiple lines of recent evidence suggest that SM and/or CE dysregulation may induce lipid peroxidation (*Choi et al., 2017*; *Tian et al., 2021*) and impinge on the activity of the proteasome and lysosomal proteases (*Gabandé-Rodríguez et al., 2014*; *van der Kant et al., 2019*). We therefore investigated different aspects of lysosomal health, including lysosomal number, size, protease activity, pH, and membrane integrity, as well as lipid peroxidation and proteasome activity, under several lipid perturbing conditions.

To monitor lysosomal protease activity, we incubated cells with BSA conjugated to self-quenched red BODIPY TR-X (DQ-Red BSA). Uptake and delivery of these molecules to the lysosome lead to BSA hydrolysis and unquenching of BODIPY fluorescence. As expected, impeding lysosomal acidification (by ATP6V1E1 KD) or lysosomal membrane integrity (by CHMP2A KD) led to reduced BODIPY

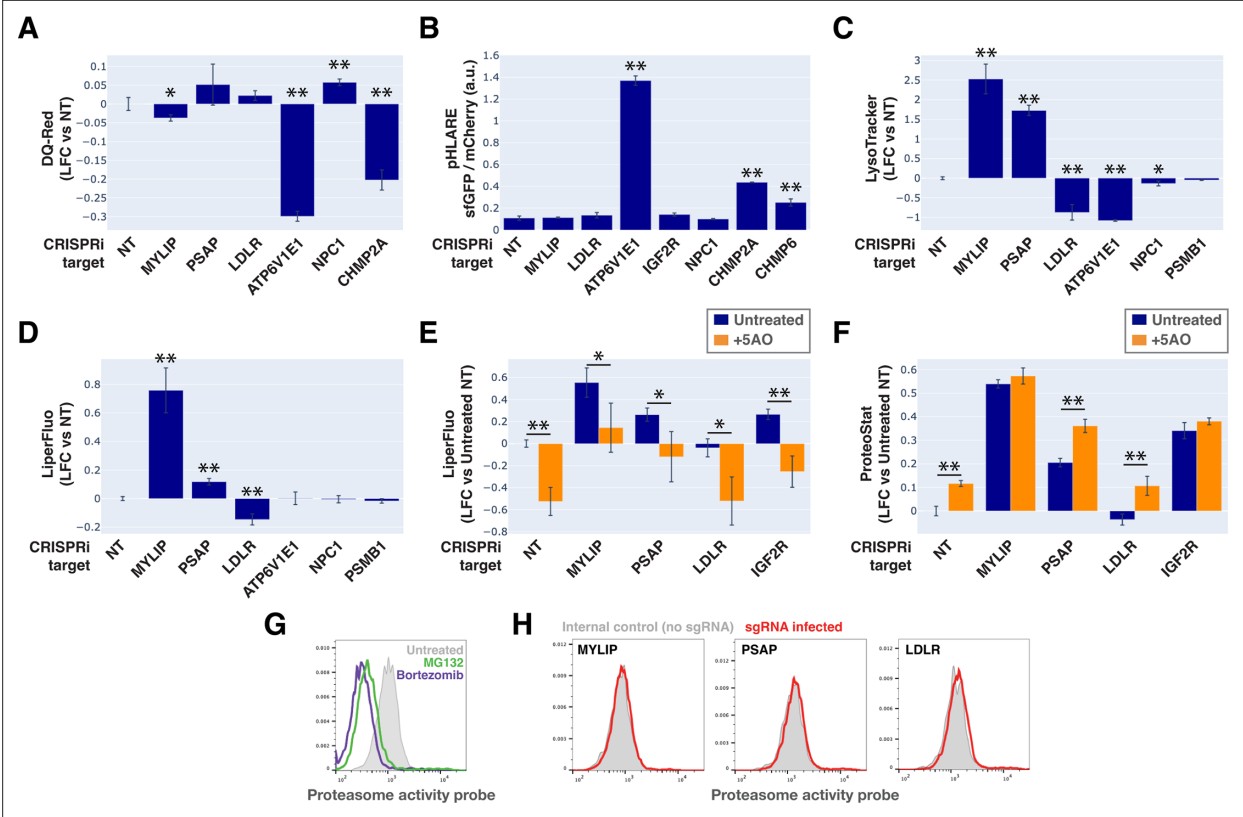

**Figure 6.** Lysosomal membrane permeabilization (LMP) contributes to proteostasis impairment. (**A–D**) Flow cytometry quantification of phenotypes in K562 cells with various KDs (n=3 for all). (**A**) TR-X fluorescence (after protease-induced unquenching) as an indicator of lysosomal protease function; (**B**) Ratio of lysosome-targeted sfGFP fluorescence (pH sensitive) to mCherry fluorescence (pH insensitive) as an indicator of lysosomal pH; (**C**) LysoTracker-Red fluorescence as an indicator of lysosomal content and pH; (**D**) LiperFluo staining as an indicator of lipid peroxidation level; (**E–F**) Efficacy of the 5 antioxidant cocktail (5AO) in rescuing the effects of gene KDs on (**E**) LiperFluo or (**F**) ProteoStat phenotypes. (✳: $p<0.05$; ✳✳: $p<0.01$; t-test vs NT control or between pairs of samples as indicated with horizontal lines; n=3) (**G**) Flow cytometry-based assay with the Proteasome Activity Probe was validated with proteasome inhibitors (n=1). (**H**) Effect of gene KDs on proteasome activity (n=1).

The online version of this article includes the following source data and figure supplement(s) for figure 6:

**Figure supplement 1.** Validation and characterization of lysosomal pH sensor and proteasome activity probe.

**Figure supplement 1—source data 1.** PDF file containing original gel image for *Figure 6—figure supplement 1D*, indicating the treatment for each lane.

**Figure supplement 1—source data 2.** Original gel image for *Figure 6—figure supplement 1D*.

unquenching (*Figure 6A*). However, despite inducing LMP to a similar degree as KDs of ATP6V1E1, CHMP2A, and CHMP6 (*Figure 5B*), lipid-modifying perturbations affected lysosomal protease function only modestly and incoherently with respect to the ProteoStat phenotype, with MYLIP KD reducing BODIPY unquenching and PSAP KD trending in the opposite direction (*Figure 6A*).

Changes in lysosomal pH (pHLys) can be detected with high sensitivity using a ratiometric reporter (pHLARE), which consists of a pH-sensitive fluorescent protein (super-folder GFP, sfGFP) and a pH-insensitive one (mCherry) flanking rat LAMP1 (*Webb et al., 2021*). Here, we stably expressed pHLARE, and validated the sensor for use with flow cytometry using a buffer series with pH ranging from 4 to 7.5 in the presence of ionophores (*Figure 6—figure supplement 1A*). The sensor also correctly reported lysosomal deacidification as expected with ATP6V1E1 KD, and demonstrated that LMP induced by either CHMP2A or CHMP6 KD was associated with modest increase in pHLys (*Figure 6B*). Nevertheless, lipid-modifying KDs – MYLIP, PSAP, and LDLR – did not significantly shift pHLys, suggesting their effects on proteostasis were not mediated via lysosomal deacidification.

We investigated the level of lipid peroxidation and lysosomal content (size and/or number) under lipid-modifying perturbations. Using a fluorescent indicator, LiperFluo, we found that perturbations that increased or decreased SM and CE levels (MYLIP and PSAP KDs, vs LDLR KD) increased or

decreased lipid peroxidation, respectively (*Figure 6D*). Interestingly, these lipid peroxidation pheno-types mirrored changes in total LysoTracker signal (*Figure 6C-D*, *Figure 6—figure supplement 1B*). As we did not observe these perturbations substantially altering pHLys (*Figure 6B*), the LysoTracker result suggested changes in lysosomal content, potentially to compensate for lysosomal impairment that may or may not be a consequence of lipid peroxidation (*Kirkegaard et al., 2010*). We tested if lipid peroxidation is necessary for the increased protein aggregation and lysosomal content using media with antioxidants. A cocktail of five antioxidants (5AO) is commonly included in the media for culturing neurons – a cell type that is particularly sensitive to oxidative stress. While 5AO was effective in reducing lipid peroxidation, it did not rescue the Proteostat or Lysotracker phenotypes (*Figure 6E–F*; *Figure 6—figure supplement 1C*). We therefore conclude that protein aggregation and lysosomal content increase are likely upstream or independent of lipid peroxidation.

Finally, we investigated if the other major arm of protein catabolism, the proteasome, is regulated by changes in lipid metabolism. A recent report suggested that high CE levels in cells may hinder phospho-Tau degradation due to proteasome inhibition (*van der Kant et al., 2019*). We labeled active proteasomes using a fluorescent probe, Me4BodipyFL-Ahx3Leu3VS (Me4BoVS; *Berkers et al., 2012*; *Berkers et al., 2007*), which covalently binds to catalytically active proteasome β-subunits in live cells. MG132 and Bortezomib treatments reduced the fluorescent signal from Me4BoVS in both live cell flow cytometry and in-gel analysis of cell lysates, validating its use as a proteasome activity probe (*Figure 6G*, *Figure 6—figure supplement 1D*). Flow cytometry analysis of CRISPRi cells showed that HMGCR KD increased proteasome activity, consistent with the reported effect of statin treatment (*Figure 6—figure supplement 1E*; *van der Kant et al., 2019*). Nevertheless, in conditions that strongly modulated SM or CE levels (KD of MYLIP, PSAP, or LDLR), we did not see significant increase or decrease of proteasome activity (*Figure 6H*). In contrast, KD of IDI1, an enzyme down-stream of HMGCR along the mevalonate pathway (but not FDFT1), strongly increased proteasome activity (*Figure 6—figure supplement 1E*). We conclude that the proteasome modulating effect of statin likely arises from changes in other metabolites along the mevalonate pathway. We summarize these results from testing the various mechanistic models in *Figure 7*.

## Discussion

Aging is associated with decreases in cellular proteostasis capacity and the accumulation of protein aggregates in diverse tissues. Despite extensive studies, factors modifying cellular proteostasis state, which has the potential to affect the state of many endogenous proteins, remain incompletely charac-terized. In this study, we performed genome-wide CRISPRi screens using a fluorescent dye sensitive to endogenous protein aggregation and uncovered links between lipid homeostasis and proteostasis. We discovered and characterized how changes in the uptake and metabolism of lipids, particularly SMs and CEs, lead to accumulation or depletion of these lipids and alter homeostasis of endogenous proteins.

SM and CE dysregulation has been implicated in various age-associated diseases. Intracellular sphingolipid accumulation has long been associated with lysosomal storage diseases, which share many features with age-related neurodegenerative diseases both at the level of physiology and cellular mechanism (*Lloyd-Evans and Haslett, 2016*). Elevated extracellular LDL cholesterol is associated with atherosclerotic cardiovascular disease (*Mortensen and Nordestgaard, 2020*), where modifications, including oxidation, and misfolding of apoB-100 can trigger the formation of Thioflavin-T-positive fibrils of aggregated LDL and stimulate uptake by macrophages and catabolism within their lysosomes (*Khoo et al., 1988*; *Parasassi et al., 2008*). Consistent with our observations, MYLIP deficiency in mice leads to upregulated LDLR expression and increased cellular uptake of lipoprotein ApoE in brain tissues, but also increased uptake and clearance of extracellular aggregated Aβ peptides (*Choi et al., 2015*). These data affirm biological connections between lipoproteins and amyloidogenesis.

What are the molecular mechanisms that impair proteostasis in response to lipid perturbation? Accumulation of sphingolipid or cholesterol due to either direct enzyme deficiency or impaired lipid transport was previously linked to reduced cellular proteostasis via impairment of lysosomal function, manifested variously as reduced lysosomal proteolytic capacity, LMP, lipid peroxidation, and aberrant mTORC1 signaling (*Kirkegaard et al., 2010*; *Gabandé-Rodríguez et al., 2014*; *Platt et al., 2012*; *Davis et al., 2021*). Surprisingly, the lipid perturbations associated with ProteoStat-positive aggre-gates did not impinge on general features of lysosomal health. More recently, Tian and colleagues

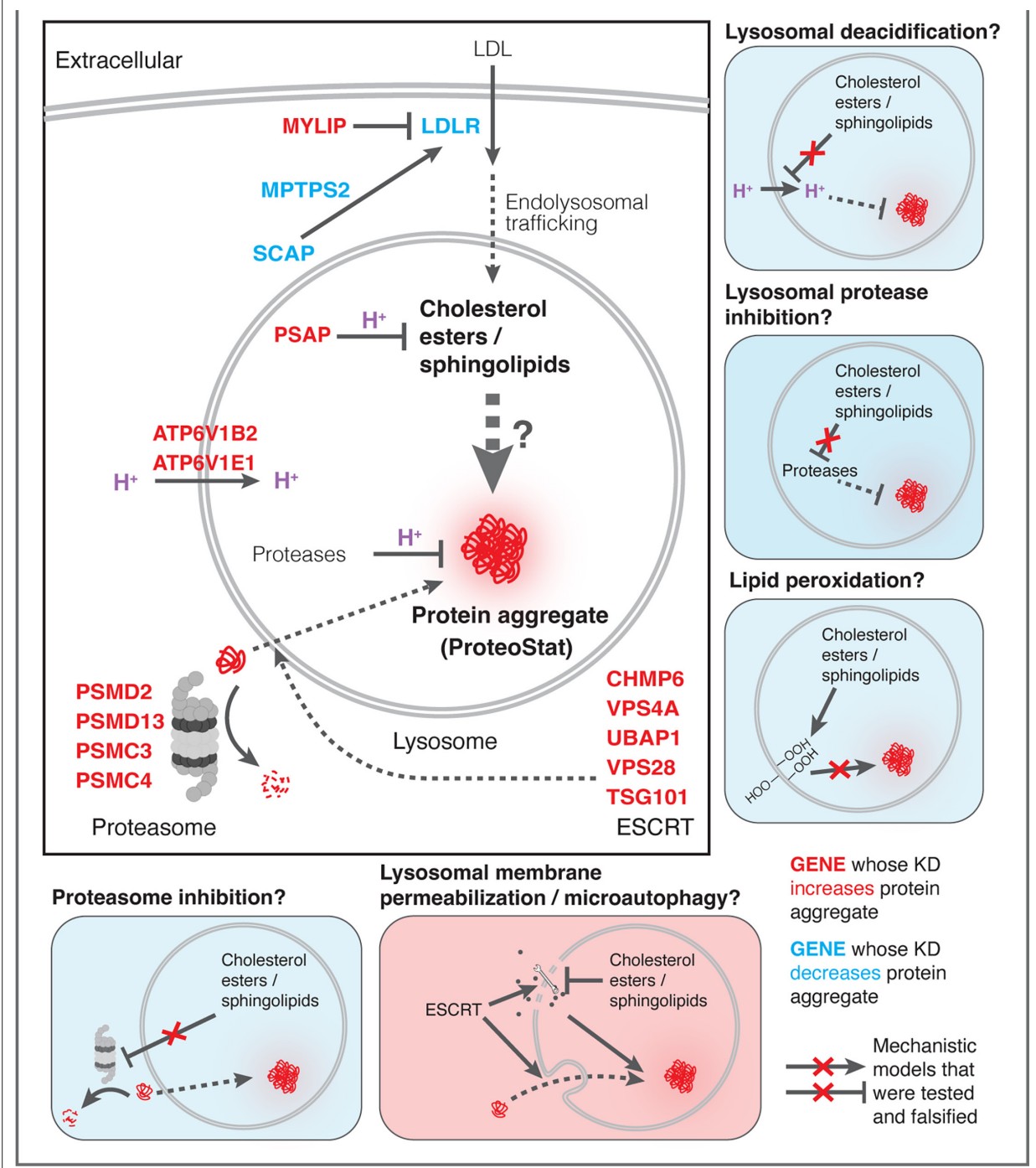

**Figure 7.** Mechanistic models linking pathways identified in CRISPRi screens and lysosomal proteomics to accrual of lysosomal protein aggregates.

discovered in iPSC-induced neurons that PSAP KD severely impairs GSL catabolism, leading to the accumulation of these lipids and, in turn, lipofuscin formation, lipid peroxidation, and eventual cell death by ferroptosis (*Tian et al., 2021*). Lipofuscins are lysosomal deposits of heterogeneous nature, consisting mainly of oxidized proteins, lipids, metal ions, and sugar residues (*Moreno-García et al., 2018*). These deposits are largely seen in post-mitotic and senescent cells, and their accumulation is observed in aged tissues but also associated with many age-related degenerative diseases (*Moreno-García et al., 2018*). PSAP KD did not induce lipofuscin in rapidly dividing K562 cells (*Figure 6—figure supplement 1F*), likely due to continuous dilution of damaged biomolecules by rapid cell

divisions before their levels can reach a threshold to form crosslinked and undegradable granules. Despite no visible accumulation of lipofuscin, we nonetheless observed an increase in lipid peroxidation and concomitant increase in lysosomal content, similarly to what's reported in iPSC-derived neurons (*Tian et al., 2021*), suggesting that granular lipofuscin formation may be dispensable for ROS generation. Furthermore, antioxidants could uncouple lipid peroxidation from lysosomal protein aggregation induced by lipid dysregulation, thus ruling out a causal relationship.

ESCRT impairment was previously linked to LMP and ESCRT-III knockdown was shown to promote the escape of exogenous tau seeds from endolysosomes (*Chen et al., 2019*). Here, we demonstrate that cholesterol and sphingolipid dysregulation can disrupt ESCRT function at the lysosome, leading to accumulation of detergent insoluble components that resist the action of VPS4A. Lipid composition might directly interfere with ESCRT function and intraluminal vesicle formation. While ESCRT-III polymerization can curve membranes and drive vesicle formation (*McCullough et al., 2015*), membrane curvature is also influenced by the shape and structure of lipid species present in the bilayer. Remarkably, enzymatic conversion of SM into ceramide (promoting neutral and negative membrane curvature, respectively) is sufficient to trigger spontaneous vesicle budding into the lumen of giant unilamellar vesicles in the absence of ESCRT (*Trajkovic et al., 2008*). Accordingly, formation of intraluminal vesicles in a mouse cell line is impaired when SM hydrolysis is inhibited (*Trajkovic et al., 2008*). Furthermore, SM hydrolysis promotes lysosomal repair in HeLa cells regardless of ESCRT status, and sphingomyelinase inhibition is detrimental to the process (*Niekamp et al., 2022*). Hence, an overabundance of SM may hinder the coalescence of ceramide-rich domains and prevent negative membrane curvature, thereby thwarting ESCRT resolution and vesicle budding. We show that this can in turn impair proteostasis and lead to aggregate accumulation, presumably as misfolded proteins are recruited to endolysosomal surfaces through microautophagy but fail to be degraded in the lumen.

Taken together, the data presented here demonstrate that, despite the general resilience of the proteostasis network, it is sensitive to pleiotropic changes in lysosomal lipid metabolism. Proteostasis is thus impacted by both changes in cell-intrinsic function and systemic lipid metabolism that can influence circulating lipids. This axis is of particular interest in the context of aging physiology in its ability to link processes that operate at the organismal and tissue levels to cellular proteostasis.

# Materials and methods

## Key resources table

| Reagent type (species) or resource | Designation | Source or reference | Identifiers | Additional information |
|---|---|---|---|---|
| cell line (*H. sapiens*) | K562 | ATCC | CCL-243 | |
| cell line (*H. sapiens*) | K562 CRISPRi | This paper | | Cell line used to perform CRISPRi screen |
| cell line (*H. sapiens*) | K562 CRISPRi +TagBFP-Gal3 | This paper | | Cell line used to assay lysosomal membrane permeabilization |
| cell line (*H. sapiens*) | K562 CRISPRi +pHLARE | This paper | | Cell line used to assay lysosomal pH |
| cell line (*H. sapiens*) | K562 CRISPRi +LysoIP | This paper | | Cell line used to perform lysosomal proteomics |
| antibody | anti-human-LAMP1 (mouse monoclonal) | Abcam | ab25630 | IF(1:50) |
| antibody | anti-mono/polyubiquitinated conjugates (mouse monoclonal) | Enzo | BML-PW8810-0100 | IF(1:1000) |
| antibody | Alexa Fluor Plus 647-conjugated anti-mouse (goat polyclonal secondary) | Invitrogen | A32728 | IF(1:1000) |
| recombinant DNA reagent | PSAP overexpression variant series | This paper | | Lentiviral constructs to interrogate the effect of individual saposins. See *Figure 2—figure supplement 2C* |
| sequence-based reagent | hCRISPRi_v2.1_top5 sgRNA library; Modified to include additional controls and target of interests | *Horlbeck et al., 2016* | | Cloned into an in-house lentiviral vector. See *Supplementary file 1* for sgRNA sequences |
| commercial assay or kit | Filipin III (in Cholesterol Assay Kit) | Abcam | ab133116 | |
| commercial assay or kit | ProteoStat Aggresome detection kit | Enzo | ENZ-51035-K100 | (1:10,000) |

*Continued on next page*

*Continued*

| Reagent type (species) or resource | Designation | Source or reference | Identifiers | Additional information |
|---|---|---|---|---|
| commercial assay or kit | Liperfluo | Dojindo Molecular Technologies | L24810 | |
| commercial assay or kit | DQ-Red BSA | Invitrogen | D12051 | |
| commercial assay or kit | LysoTracker Red DND-99 | Invitrogen | L7528 | |
| commercial assay or kit | Me4BoVS | R&D system | I-190–050 | |
| commercial assay or kit | Pierce anti-HA magnetic beads | Life Technologies | 88837 | |
| chemical compound, drug | MG132 | EMD Millipore | 474790 | |
| chemical compound, drug | Bafilomycin A1 | Invivogen | TLRL-BAF1 | |
| chemical compound, drug | VER155008 | Selleckchem | S7751 | |
| chemical compound, drug | Ganetespib | Selleckchem | S1159 | |
| chemical compound, drug | 17-DMAG | Invivogen | ant-dgl | |
| chemical compound, drug | MBCD | Sigma | C4555 | |
| chemical compound, drug | Bortezomib | Selleckchem | S1013 | |
| chemical compound, drug | LLOMe | MedKoo | 597431 | |
| chemical compound, drug | Cumene Hydroperoxide | Sigma-Aldrich | 247502 | |
| chemical compound, drug | Lipoprotein depleted FBS | Kalen Biomed | 880100 | |
| chemical compound, drug | catalase | Sigma-Aldrich | C40 | |
| chemical compound, drug | glutathione | Sigma-Aldrich | G6013 | |
| chemical compound, drug | superoxide dismutase | Sigma-Aldrich | S5395 | |
| chemical compound, drug | DL-alpha tocopherol | Sigma-Aldrich | T3251 | |
| chemical compound, drug | DL-alpha tocopherol acetate | Sigma-Aldrich | T3001 | |
| chemical compound, drug | Halt Protease and Phosphatase Inhibitor Cocktail | Thermo Scientific | 78440 | |
| software, algorithm | MAGeCK | *Li et al., 2014* | | |
| software, algorithm | GSEAPY | *Subramanian et al., 2005* | | |
| software, algorithm | MAVEN peak analysis program (version 2.20.21) | *Seitzer et al., 2022* | RRID:SCR_022491 | https://github.com/eugenemel/maven |

## Cell line construction and screen procedure

CRISPRi cell line was obtained by transducing K562 cells with a lentiviral vector expressing the nuclease-deactivated Cas9 fused with transcriptional repressor KRAB (UCOE-SFFV-KRAB-dCas9), followed by antibiotic selection. Monoclonal cell lines were obtained by limiting dilution, and screened functionally for effectiveness at silencing test genes when treated with sgRNAs. Cells were maintained in suspension at a density between 125 K to 1 M cells/mL in RPMI complete media (with 10% FBS and 1 x GlutaMax). 500 million stable CRISPRi cells were transduced with a sgRNA lentivirus pool (hCRISPRi_v2.1_top5 library, which includes 102,640 gene-targeting and 4590 non-targeting sgRNAs) at an MOI

of 0.3 for single integrants. After 6 days in culture, including antibiotic selection for cells carrying the sgRNAs with 2 µg/mL puromycin, these cells were stained with fixable blue dead cell stain (Molecular Probes), then fixed, permeabilized and stained with ProteoStat Aggresome detection kit (Enzo) according to manufacturer's protocol. Stained cells were sorted by FACS (BD FACSAria) into top and bottom quartiles (ProteoStat-high and ProteoStat-low, respectively), using gates that excluded dead cells and corrected for correlation between cell size and staining intensity. Sorted cell populations were pelleted, from which genomic DNA was extracted with phenol:chloroform:isoamyl alcohol and ethanol precipitation. sgRNA-containing sequences were amplified from the genomic DNA with two rounds of PCR, introducing multiplexing barcodes and sequencing adaptors in the process. Amplicons were then purified and sequenced with the HiSeq4000 platform (Illumina). Sequencing data were analyzed using the MAGeCK pipeline (*Li et al., 2014*). GSEA was performed using the GSEAPY package (*Subramanian et al., 2005*; *Mootha et al., 2003*) with the prerank method, using the following score that takes into similarly weighted accounts both the effect size and confidence level for each gene:

$$score_{gene} = \left( \left| LFC_{gene} / \sigma_{LFC} \right| + nlFDR_{gene} / \sigma_{nlFDR} \right) * D$$

where $nlFDR_{gene} = log_{10}\left( 1 / FDR_{gene} \right)$, and $D = 1\ or -1$ such that the overall sign matches that of $LFC_{gene}$.

## Validation of hit genes by batch-retesting a reduced pool of CRISPRi guides

A subset of sgRNAs were selected and cloned into a reduced-scale library that consisted of 572 sgRNAs targeting 286 genes (87 enriched in ProteoStat-high, 99 enriched in ProteoStat-low, and 100 unenriched but abundantly expressed in K562 cells) and 300 non-targeting sgRNAs. sgRNA transduction in K562 CRISPRi cells, ProteoStat staining, FACS, library preparation and sequencing were performed similarly to the genome-wide experiment, except at a reduced scale (20 million cells transduced with sgRNA lentivirus pool, and >2.8 million cells each of ProteoStat-high and ProteoStat-low populations were obtained by FACS).

## ProteoStat assays on cells with individual perturbations

K562 cells in multi-well plates were treated with individual perturbations and harvested for ProteoStat staining. To control for cell density-dependent variation in staining, harvested cells were spiked with untreated cells carrying appropriate fluorescent markers (e.g. TagBFP, mCerulean, or unlabeled, to be distinguished from treated cells) before fixation, permeabilization, and staining. ProteoStat fluorescence was quantified by flow cytometry (Fortessa, BD) with excitation at 488 nm and bandpass emission at 695/40 nm. ProteoStat intensity for each sample was first normalized to the internal spiked in (untreated) control, then to the corresponding negative control sample in the experiment (e.g. Non-targeting sgRNA).

Individual perturbations were achieved by CRISPRi (sgRNA lentiviruses) or pharmacological inhibitors – MG132, Bafilomycin A1, VER155008, Ganetespib, 17-DMAG, MBCD, Bortezomib, LLOMe, and Cumene Hydroperoxide. Lipid deprivation was performed by substituting regular FBS with delipidated FBS (Lipoprotein depleted FBS).

## Immunofluorescence and cell staining by fluorescent probes

Cholesterol quantification was performed by formaldehyde fixing and staining cells with Filipin III according to manufacturer's instructions.

Immunofluorescence assays were performed with standard protocol. Briefly, cells were washed with PBS and fixed with 4% formaldehyde for 30 min at room temperature, then washed with PBS. Cells were permeabilized in blocking buffer (PBS +0.5% Triton-X 100+5% goat serum) for 1 hour at room temperature, followed by incubating with blocking buffer +primary antibody at 4 °C overnight. Cells were then washed 3 times with PBS-T (PBS +0.05% v/v tween20) at room temperature and incubated with blocking buffer +secondary antibody at room temperature for 1 hr. Cells were washed twice more with PBS-T and once with PBS before imaging. If co-staining with ProteoStat, the dye was included at 1:10000 during both primary and secondary antibody incubation. List of antibodies

used: mouse-anti-LAMP1, mouse-anti-mono/polyubiquitinated conjugates, and Alexa Fluor Plus 647-conjugated goat-anti-mouse.

Lipid peroxidation was quantified by incubating cells with Liperfluo at 5 µM for 1 hr at 37 °C. Cells were resuspended in cold PBS before analysis with flow cytometry (Fortessa, BD). Antioxidants, when included, consist of the following components at the corresponding final concentration: 5 µg/mL catalase, 2 µg/mL glutathione, 15 U/mL superoxide dismutase, 2 µg/mL DL-alpha tocopherol, and 2 µg/mL DL-alpha tocopherol acetate.

Lysosomal protease activity was monitored with DQ-Red BSA. K562 CRISPRi cells carrying sgRNAs targeting corresponding genes (7 days after sgRNA introduction) were pelleted and resuspended in prewarmed trafficking media (RPMI +1% serum +1% GlutaMax +10 nM HEPES) containing 10 ug/mL DQ-Red BSA and returned to incubator at 37 °C. After incubation for 1 hr, cells were washed once with cold PBS, resuspended in cold PBS, and analyzed by flow cytometry (Fortessa, BD) with excitation at 561 nm and bandpass emission at 610/20 nm.

Lysosomal content was estimated by incubating cells under various genetic perturbations (7 days after sgRNA transduction) with LysoTracker Red DND-99 at 100 nM. After incubation at 37 °C for 1 hr, cells were washed once with cold PBS, resuspended in cold PBS, and analyzed by flow cytometry (Fortessa, BD) with excitation at 561 nm and bandpass emission at 610/20 nm.

Proteasome activity was assessed with flow cytometry by incubating cells with genetic (7 days) or chemical (1 hr) perturbations with Me4BoVS. After incubation at 37 °C for 2 hr, cells were washed once with cold PBS, resuspended in cold PBS, and analyzed by flow cytometry (Fortessa, BD) with excitation at 488 nm and bandpass emission at 525/50 nm. For in-gel fluorescence analysis, cells treated with chemical inhibitors (1–20 hr) were incubated with Me4BoVS for 1 hr at 37 °C, washed once with cold PBS and pelleted. Cells were lysed with cold NP40 lysis buffer (50 mM Tris-HCl pH7.4, 150 mM NaCl, 1% NP40, filtered) for 30 min at 4 °C and pelleted at 14,000 × $g$ for 3 min to remove debris. Samples were normalized by protein concentration (BCA assay) and run on a 4–20% Mini-PROTEAN TGX gel (Bio-Rad) and imaged directly on ChemiDoc with excitation at 460–490 nm and emission at 518–546 nm.

## Lipidomics experiment and data analysis

K562 cells under various CRISPRi or lipid perturbations were maintained, in triplicate, for 7 days. Cells were pelleted, resuspended in fresh medium, and returned to culture on Day 6 – one day before up to ~1 million cells per sample were harvested. Briefly, cells were pelleted and washed twice with PBS at 4 °C, then lysed using 800 µL of ice-cold MeOH:H2O (1:1,v/v) containing 3% by volume of lipid standards - Avanti 330707 SPLASH LIPIDOMIX Mass Spec Standard, which includes 18:1(d7) Chol Ester, d18:1-18:1(d9) SM, and Cholesterol (d7), among other lipid classes at –20 °C, and transferred to a glass vial. Following the methyl tertiary-butyl ether (MTBE)- liquid-liquid extraction (LLE) adapted from the Matyash protocol (*Matyash et al., 2008*), 800 µL of 100% MTBE was added into each sample, vortexed, and incubated on ice for 10 min to form phase separation. Next, after centrifuging samples at 3000 × $g$ at 4 °C for 5 min, 500 µL of the upper organic phase was collected into a clean glass vial, and the lower phase was re-extracted by the addition of 600 µL of 100% MTBE as described above. The organic phases were combined, dried under nitrogen, and resuspended in 150 µL of HPLC graded-ButOH/MeOH/H2O (2:2:1,v/v/v) with internal standards including: 5 µg/mL PE(18:0/18:0)-D70, PA(14:0/14:0)-D54, and 25 µg/mL LPC (16:0)-D31. Samples were analyzed using Thermo Scientific Vanquish UHPLC coupled with Q Exactive Plus Mass Spectrometer with 7.5 µL injection volume.

Analysis via separate injections in negative and positive mode ionization were done on a Vanquish UPLC coupled to Q-Exactive Plus mass spectrometers (Thermo Fisher Scientific). Lipids were separated on a Thermo Accucore C30 (250x2.1 mm, 2.6 µm polymer particles). Mobile phase A was 20 mM ammonium formate in MeCN:water (60:40,v/v); mobile phase B was 20 mM ammonium formate in isopropanol:MeCN (90:10, v/v) and a flow rate of 0.2 mL/min was used. The column was equilibrated for 7 min in 30% B prior to injection; followed by a gradient of 30–43%B from 0 to 7 min, 43–65% B from 7 to 12 min, 65–70% B from 12 to 30 min, 88% B at 31 min, 88–95% B at 51 min, 100% B at 53 min and 55 min, and 30%B from 55 to 60 min.

The source parameters for the MS were as follows: Sheath gas: 40; Aux gas:15; Sweep gas: 1; spray voltage 3.1 kV, the capillary temperature of 275 °C, and S-lens RF level of 50. The mass spectrometer

was operated in data-dependent top-8 MS2 mode, with 14000 resolution setting and AGC of 3e6 for MS1, and 17,500 resolution setting and AGC target of 3e6 for MS2. Stepped, normalized collision energies of 20, 30, 40 were used.

Data were analyzed and visualized using the open source software MAVEN peak analysis program (https://github.com/eugenemel/maven [version 2.20.21, RRID:SCR_022491]; *Seitzer et al., 2022*; *Seitzer and Ledogar, 2024a*). Compounds were identified with the criteria of a precursor ion tolerance of 10 ppm and a product ion tolerance of 20 ppm, comparing fragmentation and retention time to in-house generated in-silico libraries for lipidomics (https://github.com/calico/CalicoLipidLibrary) (*Seitzer and Ledogar, 2024b*).

## PSAP variant series and rescue experiments

A series of constructs expressing various PSAP variants were built into lentiviral vectors under a UBC promoter. The canonical sequence mRNA variant 1 (NM_002778.4) was designated as wild-type. The full-length, wild-type sequence was modified using the following known disease-associated mutations to generate deficient variants: SapA-deficiency (p.V70del), SapB-deficiency (p.N215H_C241S), SapC-deficiency (p.L349P_C388F). The SapD-deficiency variant was created by truncating the entire C-terminal starting at the first SapD residue (p.405_524del). These mutations are procured from OMIM (https://omim.org/entry/176801) and other sources in literature (*Spiegel et al., 2005*; *Schnabel et al., 1991*).

PSAP KD rescue experiments were performed by co-transducing K562 CRISPRi cells with lentiviruses expressing either a PSAP-targeting or non-targeting sgRNA, and in parallel one of the various PSAP variants. Transduced cells were selected with 2 µg/mL puromycin and 500 µg/mL geneticin for doubly transduced cells. Cells were harvested 6 days after transduction, spiked with untreated cells as internal control, then stained with ProteoStat and analyzed with flow cytometry as in other experiments.

## Lyso-IP, silver stain, differential extraction

Lyso-IP cells were obtained by transducing K562 CRISPRi cells with a lentiviral vector expressing a lysosomal membrane protein fused with an epitope and fluorescent protein (UCOE-SFFV-TMEM192-mRFP1-3xHA-2A-Neo), followed by antibiotic selection. Cells were transduced in triplicates with corresponding sgRNA lentivirus and expanded in culture for 6 days, including antibiotic selection with 2 µg/mL puromycin for cells carrying the sgRNAs. Lysosomes were harvested in a protocol partly adapted from the Sabatini Lab (*Abu-Remaileh et al., 2017*). Briefly, up to 40 million cells per sample were harvested, processing quickly on ice or at 4 °C in batches of four to six samples. Cells were washed twice with chilled 10–25 mL PBS by pelleting at 1000 × *g* for 2 min each, followed by resuspension in 1 mL KPBS (136 mM KCl, 10 mM $KH_2PO_4$, adjusted to pH 7.25 with KOH) and a final centrifugation in an 1.5 mL low-retention Eppendorf tube. Cell pellets were resuspended in 950 µL KPBS +inhibitors (Halt Protease and Phosphatase Inhibitor Cocktail (from 100 x)+0.5 mM PMSF). One at a time, cell suspensions were passed 20 times through a cell homogenizer (Isobiotec) with a 8 µm clearance. Cell homogenates were clarified by pelleting at 1000 × *g* for 2 min to remove cell debris and nuclei. Clarified supernatants were incubated with 100 µL of KPBS-prewashed anti-HA magnetic beads on rotation for 3 min and put on a DynaMag magnet rack. Lysosomes bound on beads were washed thrice with cold 1 mL KPBS directly on the magnet. After final buffer removal, washed beads were flash-frozen and stored at –80 °C until lysis, when they were thawed briefly on ice.

Bound lysosomes were lysed by incubating in up to 150 µL (scaled with cell number) 1 x RIPA complete buffer (RIPA (from Abcam 10 x)+2 x Halt Protease and Phosphatase Inhibitor Cocktail (from 100 x)+0.5 mM PMSF +Benzonase (from 1000 x)+2 mM MgCl2) on rotation for 30 min. Beads were then removed with magnets and then again by centrifuging at 1000 × *g* for 2 min. An aliquot of this Lysosomal Total fraction was set aside, and the rest is pelleted at 21,000 × *g* for 30 min to separate into the Lysosomal Soluble fraction (supernatant) and Lysosomal Insoluble fraction (pellet). The RIPA-insoluble pellets were washed 1 x with 100 µL 1 x RIPA complete buffer with brief vortexing (10 s), and pelleted again at 21,000 × *g* for 30 min. Final pellets were resuspended with up to 80 µL (scaled with cell number) 1 x Urea/SDS complete buffer (8 M Urea, 2% SDS, 50 mM DTT, 50 mM Tris-HCl, 1 x Halt Protease and Phosphatase Inhibitor Cocktail (from 100 x), pH 7.4), and bath-sonicated at low power for 10 min. All fractions were flash-frozen and stored at –80 °C until further analysis.

For polyacrylamide gel and silver stain analysis, 30 µL per Lysosomal Insoluble fractions were loaded onto each lane of a 4–20% Mini-PROTEAN TGX gel (Bio-Rad) and stained with Pierce Silver Stain Kit, and imaged on the Odyssey CLx Imager (LI-COR) according to manufacturer's protocols.

## Protein extraction and TMT labeling for proteomic analysis

Lysosomal Total (in 1 x RIPA complete buffer) and Lysosomal Insoluble (in 1 x Urea/SDS complete buffer) samples (see above section for protocol and buffer compositions) were prepped using a combined manual version of our in-house automated multiplexed proteome profiling platform (AutoMP3) protocol (*Gaun et al., 2021*) and the single-pot, solid-phase-enhanced sample-preparation protocol (*Hughes et al., 2019*) with some minor modifications. In brief, 30 µL of the samples were diluted with 50 µL PBS. Samples were heated to 95 °C for 5 min, briefly cooled and then treated with benzonase for 30 min at 37°C followed by sonication for 5 min. Samples were reduced with 5 mM DTT at 37°C for 30 min, alkylated with 15 mM iodoacetamide for 20 min in the dark and then quenched with 5 mM DTT for 15 min at room temperature in the dark. Protein cleanup was performed using a 1:1 mixture of E3:E7 Sera-Mag Carboxylate-Modified Magnetic Particles (Cytiva Life Sciences, Marlborough, MA, USA). In brief, a bead stock was made and first washed with water four times, and then diluted to (20 µg solids/µL) and 100 µL beads were taken out into a new tube, placed on a magnet and the water removed followed by the addition of 100 µL of the sample to the beads and then mixed thoroughly off the magnet. The mixture was then immediately brought to 75% acetonitrile by addition of 300 µL of 100% acetonitrile and then incubated 18 min at room temp, allowing beads to aggregate with proteins. Tubes were then immobilized on the magnet and the supernatant removed. The beads were washed twice with 70% ethanol, and once more with 100% acetonitrile before they were brought into suspension with 92 µL digestion buffer (50 mM EPPS pH 8.5, 10 mM CaCl2) and Trypsin/LysC (Promega) was added at a ratio of 1:25 (enzyme: substrate) and incubated for 16 hr at 37 °C with shaking (800 rpm). The next day vials were sonicated for 5 min and tubes were then immobilized on the magnet and the digest transferred to new vials. For TMT-labeling 40 µL of the digest was then mixed with 12 µL TMTpro-reagent (0.2 mg/ml, in acetonitrile) followed by incubation at 25°C for 1 hr. All samples were quenched with 11 µL 5% hydroxylamine solution before they were combined. The combined samples were mixed and split into multiple peptide cleanup reactions in order to be able to reach the high % of needed organic solutions for peptide binding to beads. In brief, 50 µL prewashed beads were mixed with 50 µL of the combined sample and then the solution was immediately brought to 95% isopropanol by addition of 100% isopropanol. These samples were incubated at room temperature for 18 min to enable peptide binding to the beads. Tubes were then placed on the magnet for 2 min and the supernatant removed followed by two rounds of washes with 95% isopropanol, and then once with 100% acetonitrile. The peptides were then eluted in two rounds of 40 µL 5% acetonitrile followed by shaking at 1500 rpm for 1 min. The cleaned up TMT-labeled samples were dried down in a speedvac (Labconco) and then resuspended in 5% formic acid, 5% acetonitrile (targeting 0.25 µg/µL) and 4 µL was injected for analysis per run.

## Mass spectrometry proteomic analysis

All data were obtained on an Orbitrap Eclipse mass spectrometer with FAIMS (Field Asymmetric Ion Mobility Spectrometry) Pro Interface. The mass spectrometer was coupled to an UltiMate 3000 HPLC operating in DDA mode (Thermo Fisher Scientific). Peptides were separated on an Aurora Series emitter column (25 cm ×75 µm i.d., 1.6 µm, 120 Å pore size, C18; IonOptics) using a 165 min gradient from 8 to 30% acetonitrile in 0.125% formic acid. FAIMS was enabled during data acquisition with compensation voltages set as −40, −50, −60, and −70 and each voltage had a cycle time of 1.25 s. A high-resolution MS1 scan was performed in the Orbitrap at a 120,000 resolving power, m/z range 400–1600, RF lens 30%, standard automatic gain control (AGC) target, and 'Auto' maximum injection time and data collected in profile mode. For MS2 collision-induced dissociation (CID) was used and the fragments were analyzed in the ion trap. A 0.7 m/z isolation window was used and the normalized collision energy for CID was 35% with a CID activation time of 10ms at 0.25 activation Q and AGC $1\times10^4$ with 35ms maximum injection time and data collected in centroid mode. Real-time search (RTS; *Schweppe et al., 2020*) with the Uniprot database UP000005640 was used. Depending on the RTS, the ions were then analyzed with SPS MS3. For the RTS, the minimum Xcorr was set to 1, the minimum dCn was set to 0.1, and the maximum missed cleavages allowed was set to two. MS3

analysis was performed in the Orbitrap at a 50,000 resolving power, with HCD activation and 45% normalized collision energy, AGC 1×105, m/z range 100–500 m/z, and 200ms maximum injection time and data collected in centroid mode. A maximum of 10 fragment ions from each MS2 spectrum were selected for MS3 analysis using SPS.

## Computational interpretation of proteomic Data

An in-house software pipeline (version 3.12) was used to process all mass spectrometry data (*Huttlin et al., 2010*). Raw files were first converted to mzXML files using the MSConvert program (version 3.0.45) to generate peak lists from the RAW data files, and spectra were then assigned to peptides using the SEQUEST (version 28.12) algorithm (*Eng et al., 1994*). Spectra were queried against a 'target-decoy' protein sequence database consisting of human proteins (using the reviewed Swiss-Prot portion of UP000005640 containing 20596 features and common contaminants) in forward and reversed decoys of the above (*Elias and Gygi, 2007*). The parent mass error tolerance was set to 20 ppm and the fragment mass error tolerance to 0.6 Da. Trypsin specificity was required allowing for up to two missed cleavages. Carbamidomethylation of cysteine (+57.02 Da), TMTpro-labeled N terminus and lysine (+304.20 Da) were set as static modifications. Methionine oxidation (+15.99 Da) were set as variable modifications. Following database searching, linear discriminant analysis was performed to filter peptide spectral matches to a 1% false discovery rate (FDR; *Huttlin et al., 2010*). Following peptide filtering, non-unique peptides were assigned to proteins that comprised the largest number of matched redundant peptide sequences using the principle of Occam's razor. The quantification of TMT reporter ions was performed by extracting the most intense ion at the predicted m/z value for each reporter ion (within a 0.003 m/z window) and isotopic purity correction of reporter quant values was applied. Known false positives (i.e. decoys and contaminants) were excluded from further analysis steps and peptide intensities and signal-to-noise ratios were exported for further analysis. In order to identify differentially expressed proteins between each condition the msTrawler v1 statistical software package (*O'Brien et al., 2024*) was used.

## Lysosomal membrane permeability assay

Stable LMP reporter cells were generated by transducing K562 CRISPRi cells with a lentiviral vector expressing TagBFP-GAL3, followed by selection with 10 μg/mL blasticidin. To measure the effect of gene KDs on LMP, these reporter cells were transduced with corresponding sgRNA-expressing lentiviruses and selected with 2 μg/mL puromycin. 5 days after sgRNA transduction, cells were plated on poly-D-lysine-coated CellCarrier Ultra 96-well plates (Perkin Elmer) at 12 K cells in 200 μL regular media per well. 2 days after plating, cells were treated with 60 uM LLOMe (or left untreated) and imaged on Opera Phenix imaging platform (Perkin Elmer) at 15 minute intervals for 4 hr, with excitation laser at 375 nm and bandpass emission filter at 435–550 nm. Images were analyzed with Harmony software (Perkin Elmer) by cell segmentation and spot detection. The number of GAL3 puncta in each field of view (6 fields per condition) is normalized by the total segmented cell area and scaled to match the average cell area, to correct for cell density and averaged across replicates. The maximal response for each sample over the 4 hr time course was reported.

## pHLys quantification by pHLARE

Stable pHLys reporter cells were generated by transducing K562 CRISPRi cells with a lentiviral vector expressing pHLARE (a fusion protein consisting of prolactin signal sequence, superfolder GFP [sfGFP], rat LAMP1, and mCherry developed by *Webb et al., 2021*, and selected with 10 μg/mL blasticidin.

For validation, reporter cells were pelleted at 500 × *g* for 5 min and resuspended in a buffer series at pHs ranging from 4 to 7.5 (140 mM KCl, 1 mM MgCl2, 1 mM CaCl2, 5 mM glucose and 10% citrate-phosphate-borate buffer at the target pH; *Carmody, 1961*) in the presence of ionophores (10 μM nigericin, 2 μM monensin). Non-reporter cells were treated in parallel to correct for changes in background autofluorescence under various perturbations. After incubation at 37 °C for 10 min, cells were analyzed with flow cytometry (Fortessa, BD), using excitation-bandpass emission settings of 488 nm-525/50 nm and 561 nm-610/20 nm for sfGFP and mCherry, respectively. To elucidate the effect of gene KDs on pHLys, CRISPRi cells with or without stable pHLys reporter were transduced with sgRNA-expressing lentiviruses, selected with 2 μg/mL puromycin, and analyzed similarly with flow cytometry, 7 days after sgRNA transduction.

Cytometry data was analyzed in FlowJo software (BD). Median background fluorescence in each channel in the cognate non-reporter controls was subtracted from the sfGFP and mCherry fluorescence intensities in the pHLys reporter samples. The ratio of adjusted sfGFP intensity vs adjusted mCherry intensity was reported.

## Giant unilamellar vesicles (GUVs) and ProteoStat staining

GUVs with binary mixtures of saturated and unsaturated lipids were obtained by electroformation. Briefly, DOPC, DPPC, and free cholesterol were mixed in equal molar parts to a final concentration of 1 mg/mL. TopFluor Cholesterol (Avanti Polar Lipids Inc) was included at 1% of total lipids. The lipid mix was spread onto ITO-coated glass coverslips and dried overnight under vacuum. Dried lipids were hydrated with 300–400 ul of a 100 mM sucrose solution while applying a sinusoidal voltage (1.2 Vp-p and 10 Hz) for 1 hr at 60 °C. GUVs were harvested, diluted 20-fold in 200 µL of a 100 mM glucose solution supplemented with either no ProteoStat, or ProteoStat at a final dilution of 10,000 x and 500 x and imaged at 30 °C in a spinning disk confocal fluorescence microscope (Nikon), using excitation/emission at 488 nm/525±18 nm (TopFluor) or 561 nm/605±35 nm (ProteoStat). Multi-channel single-plane images were acquired using NIS-Elements software and analyzed with ImageJ/Fiji (*Schindelin et al., 2012*).

## Transmission electron microscopy

K562 CRISPRi cells carrying either non-targeting or PSAP-targeting sgRNA (6 days after sgRNA transduction) were fixed by directly mixing cells in suspension 1:1 with prewarmed 1 x fixative (2% glutaraldehyde and 4% paraformaldehyde in 0.1 M sodium cacodylate buffer, pH7.4), followed immediately by gentle pelleting at $100 \times g$ for 10 min. Cell pellets were resuspended gently with fresh 1 x fixative, and incubated at room temperature for 1 hr, followed by washing once with 2 mL 0.1 M sodium cacodylate buffer at room temperature and then transferring to 0.5 mL 0.1 M sodium cacodylate buffer, each time pelleting at $100 \times g$ for 10 min.

Fixed cells were stored at 4 °C until they were enrobed in gelatin and post-fixed with 1% osmium tetroxide, washed with ddH$_2$O and stained with 1% uranyl acetate. Stained samples were dehydrated in a graded series of ethanol, infiltrated and embedded in epon resin. Samples were then trimmed and sectioned with an ultramicrotome and imaged on a JEM-1400 transmission electron microscope (JEOL).

## Materials availability

All new CRISPRi and reporter cell lines, and the pooled sgRNA plasmid library are available from the corresponding author.

## Data and availability

Number of replicates (n), where indicated, refers to biological replicates, that is samples that are independently treated and processed. Standard and publicly available MAGeCK and GSEAPY packages were used to analyze the CRISPRi screen data. The mass spectrometry proteomics data have been deposited to the ProteomeXchange Consortium via the PRIDE (*Perez-Riverol et al., 2022*) partner repository with the dataset identifier PXD054648.

## Acknowledgements

We thank Jonathan Paw for flow cytometry and cell sorting; Margaret Roy, Andrea Ireland, Nelda Yi, Irene Lam, Twaritha Vijay, and Nicole Fong for Illumina sequencing; David Harris, Dan Gottschling, and Cynthia Kenyon for critical reading of the manuscript. TEM was performed by Stanford University Electron Microscopy.

## Additional information

### Competing interests

John Yong, Jacqueline E Villalta, Ngoc Vu, Niclas Olsson, Magdalena Preciado López, Julia R Lazzari-Dean, Kayley Hake, Fiona E McAllister, Bryson D Bennett, Calvin H Jan: Employee of Calico Life Sciences LLC. Matthew A Kukurugya: Former employee of Calico Life Sciences LLC.

### Funding
No external funding was received for this work.

### Author contributions
John Yong, Conceptualization, Data curation, Formal analysis, Investigation, Visualization, Methodology, Writing – original draft, Writing – review and editing; Jacqueline E Villalta, Kayley Hake, Resources, Methodology; Ngoc Vu, Matthew A Kukurugya, Data curation, Formal analysis, Investigation, Writing – review and editing; Niclas Olsson, Data curation, Formal analysis, Investigation, Methodology, Writing – review and editing; Magdalena Preciado López, Investigation, Writing – review and editing; Julia R Lazzari-Dean, Resources, Methodology, Writing – review and editing; Fiona E McAllister, Bryson D Bennett, Supervision, Writing – review and editing; Calvin H Jan, Conceptualization, Formal analysis, Supervision, Writing – original draft, Writing – review and editing

### Author ORCIDs
John Yong ⓘ https://orcid.org/0000-0002-4914-2259
Julia R Lazzari-Dean ⓘ https://orcid.org/0000-0003-2971-5379
Calvin H Jan ⓘ https://orcid.org/0000-0001-9033-4028

Reviewer #1 (Public Review): https://doi.org/10.7554/eLife.86194.3.sa1
Reviewer #2 (Public Review): https://doi.org/10.7554/eLife.86194.3.sa2
Author response https://doi.org/10.7554/eLife.86194.3.sa3

---

# Additional files

### Supplementary files
• Supplementary file 1. hCRISPRi_v2.1_top5 sgRNA library. Modified from *Horlbeck et al., 2016* to include additional controls and target of interests.

• MDAR checklist

### Data availability
Source data files for CRISPRi screens, lipidomics, and proteomics in this study are included in the manuscript and supporting files.

The following dataset was generated:

| Author(s) | Year | Dataset title | Dataset URL | Database and Identifier |
|---|---|---|---|---|
| Yong J, Villalta JE, Vu N, Kukurugya MA, Olsson N, López MP, Lazzari-Dean JR, Hake K, McAllister F, Bennett BD, Jan CH | 2024 | Impairment of lipid homeostasis causes lysosomal accumulation of endogenous protein aggregates through ESCRT disruption | https://www.ebi.ac.uk/pride/archive/projects/PXD054648 | PRIDE, PXD054648 |

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
