## [Editor Report · eLife Assessment]

Protein and lipid homeostasis is essential for maintaining cellular functions but their crosstalk remains largely unknown. This **important** manuscript deals with this interesting topic and applies the powerful unbiased tools of somatic cell genetics to discover evidence suggesting a link between sphingolipids/cholesterol ester metabolism and lysosomal protein aggregation. The authors provide **compelling** orthogonal evidence to support their conclusions.

---

## [Referee Report · Reviewer #1 (Public Review)]

In this manuscript, Yong and colleagues link perturbations in lysosomal lipid metabolism with the generation of protein aggregates resulting from proteosome inhibition. The main tool used is the ProteoStat stain to assess protein aggregate burden in native cells (i.e. cells under no exogenous or endogenous stress). They initially use CRISPR-based genome-wide screens to identify several genes that affect this aggregate burden. Interestingly, knockdown of genes involved in lysosomal acidification was a major signature which led to identification of other culprit lysosome-associated genes that included ones involved in lipid metabolism. Subsequent CRISPR screen focused on lipidomic analysis led to identification of sphingolipid and cholesterol esters as lipid classes with effects on proteostasis.

Comments on revised version:

They did a decent job addressing most of my comments and the new data (including LysoIP) makes for much more plausible conclusions.

They propose the idea that microautophagy is mediating the delivery of these aggregates to lysosomes.

It appears there are enough experiments and support now for their premise.

The lysosomal lipid metabolism link to proteostasis is still a lingering question in this work but they addressed each of the points I raised regarding it and revised the manuscript accordingly with pertinent discussion.

It is difficult to truly address the lipid link and I think we have to acknowledge that. But overall, looking at the effort and conclusions, this has been improved enough to be a valuable contribution to the field.

---

## [Referee Report · Reviewer #2 (Public Review)]

In this paper, starting with unbiased CRISPRi screening, the authors found that perturbations in lipid homeostasis lead to proteostasis impairment. The screen and most follow-up experiments used the dye ProteoStat, which detects protein aggregates and the aggresome. Based upon their screen hits and subsequent analyses, the authors determined that increased levels of sphingolipids and cholesterol esters induce proteostasis defects, along with formation of protein aggregates that appear to be localized in the lysosome. The lysosome increases in content, but its function is not detectably perturbed.

Comments on revised version:

I am satisfied with the authors' actions in response to my public and specific suggestions, but not yet with the manuscript itself. I think that the paper would be improved if they showed the evidence arguing against an effect on proteasome activity but I can live with this omission. I think that the readability and ease of grasping the main points are improved by Figure 7. Inclusion of these simple but informative conceptual summaries is a must.

---

## [Author Response]

We are submitting a revised manuscript with major additions that address the main concerns in the initial reviews. At the highest level, this revision provides (i) orthogonal biochemical measurements that yield concrete evidence of lysosomal protein aggregates, and (ii) a plausible mechanism linking lysosomal lipid handling and protein aggregation through disruption of ESCRT function. We believe these additions significantly improve the completeness of this study and the conclusions that can be drawn from the data.

Below are more specific highlights on the addition in this revision:

- We included orthogonal techniques (thioflavin-T staining and Lyso-IP followed by differential extraction) and confirmed the accumulation of RIPA-insoluble protein aggregates at the lysosomes in cells under lipid perturbation (Figure 3).

- We performed TMT-Proteomics and identified accumulation of insoluble ESCRT components at the lysosomes under lipid perturbation (Figure 4). Two new authors involved in this effort are added onto the manuscript.

- The ESCRT result prompted us to revisit lysosomal membrane integrity. With improved imaging conditions and analysis we were able to see increased membrane permeabilization under lipid perturbation. VPS4A overexpression partially rescued this phenotype, suggesting that lipid accumulation impairs ESCRT disassembly (Figure 5).

- Together, the results suggest that lipid perturbation impairs ESCRT function, compromising both lysosomal membrane repair and microautophagy, resulting in the accumulation of endogenous protein aggregates at the lysosomes (Graphical Abstract).

**Reviewer #1 (Recommendations For The Authors):**
(1) Perhaps the most prominent limitation of this work is the unilateral focus on native cells (i.e. cells under no endogenous or exogenous stress) as the model for protein aggregate formation. Furthermore, although the ProteoStat stain has been utilized by many investigators before, the sole reliance on this stain as the read-out for their assays is concerning. To compound the concern, the ProteoStat-positive puncta co-localize with lysosmal markers which was surprising even to the authors. All in all, it behooves the authors to test proteostasis in multiple parallel ways to actually define what they are studying. How is it possible that protein aggregates under native conditions are only co-localized with lysosomes? Are we really studying protein aggregates which should predominantly be cytoplasmic insoluble aggregates?(a) They need to get away from a simple stain like ProteoStat and conduct co-stainings with other markers such as poly-ubiquitin antibodies and other chaperones to define what and where else exactly are these aggregates.

Co-staining with poly-ubiquitin was included in the original manuscript. We added orthogonal staining with another widely used amyloid dye, Thioflavin-T, and provided fine-grained quantification of lysosomal vs cytosolic localization of various signals (Figures S4A-C & 3A-B).

(b) They need to do Immunoblots with and without triton insolubility to see if these aggregates are insoluble as most would predict. They can do lysosomal isolation vs cytoplasmic to see if the insoluble aggregates are really lysosomal.

We performed Lyso-IP followed by differential detergent extraction to confirm the accumulation of insoluble proteins at the lysosomes (Figure 3C). Proteomic analysis identified some of these insoluble proteins as ESCRT subunits (Figure 4).

(c) They should compare aggregate formation in the native state versus cells with lysosomal inhibition via Bafilomycin or chloroquine versus cells with proteosomal inhibition. The lysosomal inhibition experiments are particularly informative given the lysosomal relevance they have uncovered.

We included other small molecule inhibitors and at different time points to compare the effect of different modes of proteostasis challenge (Figure S4A-D). Together with the ESCRT finding, our results suggest the role of microautophagy in our system, and provide a model of how ProteoStat- and/or ubiquitin- positive substrates become partitioned between the cytoplasm and lysosomes under different perturbations.

(d) Many protein aggregates which are too bulky for proteosome degradation will traditionally be dealt with by aggrephagy. Why is this not observed?

Knockdown of core macroautophagy components did not impact Proteostat intensity in our CRISPRi screen, suggesting that basal macroautophagy plays a negligible role in clearing endogenous amyloid-like structures in our experimental system. We provide an alternative model that these aggregates instead arrive at the lysosomes via microautophagy.

(2) After addressing #1, they can validate if the genes they identified by CRISPR screens are also important in modulation of protein aggregate burden in other systems. For example, if they inhibit lysosomes by Bafilo or Chloroquine to obtain protein aggregates and then Knockdown the identified genes in the CRISPR screens, will they get the same results?

We addressed the effect of different modes of proteostasis challenge as recommended above. Deacidifying the lysosomes alone causes intense protein aggregation (Figure S4A-D) and eventually cell death, and was thus not combined with other perturbations.

(3) They identify lysosomal lipid metabolism genes/pathways as the culprit for inducing proteostasis. In particular sphingolipid and cholesteryl ester species appear to be operational here. However, there are no specific lipids species or specific lipid metabolism gene that is causative. Rather, you have to knockdown entire processes to have an effect. This suggests that the focus on lysosome health (i.e. permeability, proteolysis, etc) is rudimentary. When you have to knockdown entire classes of lipids, this would indicate more broad effects on cellular lipids (including membrane lipids beyond the lysosome) and related cellular health?

We included data on the effect of knocking down MYLIP, PSAP, and as a comparison PSMD2 on the growth rate of K562 cells (Figure S5A). MYLIP and PSAP KDs, which cause predominantly an accumulation of lipids, do not impede cell growth. Increasing lipid uptake by MYLIP KD increases cell proliferation under our culture conditions, suggesting a general negative impact on cell health was not required for the association between lipid levels and protein aggregates.

(a) They conduct a superficial methyl-beta-cyclodextrin experiment with equivocal results. The use of MBCD for different time-courses to deplete various membrane cholesterol pools including the plasma membrane pool is important to ascertain what aspect of the cellular cholesterol is affecting proteostasis. MBCD +/- cholesterol reintroduction time-courses for rescue will also be key to determine the culprit cellular cholesterol pool.

The MBCD / Filipin experiment helped us determine that ProteoStat doesn’t directly stain cholesterol, nor any major plasma membrane components. Free cholesterol was implicated in neither the screen nor the lipidomics and was not the subject of targeted experiments.

(b) The same concept can be applied to sphingolipids. There are sphingolipids in abundance in multiple membrane compartments. Which ones are causal here? More nuanced evaluation of this with sphingolipid staining/tracking can be conducted.

We attempted experiments where sphingolipids were added back to cells grown in FBS-depleted media. Nevertheless, we were not able to consistently deliver these lipid species and doing so while ensuring the correct subcellular localization at physiologically relevant level would require substantial methods development.

(c) As part of this, are lipid rafts and/or caveolae being affected by the perturbations in cholesterol and sphingolipids? Lipid rafts are highly enriched in these 2 lipids which could link to their preteostasis observation.

Indeed, ceramides released from SM hydrolysis are proposed to self-assembled into microdomains with negative curvature that can promote the formation of intralumenal vesicles (Alonso and Goni, 2018; Niekamp et al 2022). We propose that SM accumulation may hinder this process by counteracting the negative membrane curvature and impede microautophagy.

(d) How about ER membrane lipids? The UPR and subsequent effects on proteostasis are intricately involved with ER lipid bilayer composition.

We did not perform lipidomics on ER membranes in this study, though we note that at steady state, sphingolipids and cholesterol esters are not expected to be enriched at the ER (Ikonen and Zhou, 2021). We checked whether lipid-related genetic perturbations induced the UPR in published perturb-seq data in K562 cells. Neither MYLIP nor PSAP knockdown induced a UPR.

In conclusion, the manuscript is interesting but the excitement over a link between lysosome-related lipid metabolism and proteostasis needs to be tamped until a more robust experimental approach is employed to generate supportive and corroborating results.
**Reviewer #2 (Recommendations For The Authors):**
- The paper has a number of grammatically awkward sentences. Editing these would enhance clarity.- It is important to show the co-localization of aggregates with the lysosome. This is shown in supplements but should be in a main figure. Here the authors cite previous work indicating that ProteoStat puncta co-localize with ubiquitinated proteins and state that they do not see this, then essentially just move on. Is there an explanation for this discrepancy and can it be resolved? What do they think is really going on? What happens to levels of ubiquitinated proteins when lipid metabolism is perturbed as in these experiments?

We have included the lipid-induced lysosomal protein aggregation data in the main text (Figure 3A-B), and provided fine-grained quantification of the cytosolic-vs-lysosomal ProteoStat / Ub / ThT signals under different aggregate-inducing conditions (Figure S4A-D). We discuss these results in the main text and propose a model involving ESCRT-mediated microautophagy in the main text. This is supported further by the LysoIP-proteomics and LMP analysis.

- Please add an indicator of amino acid numbers to Fig. 3C.

These annotations are now included (now Figure S3C).

- The legend for 3D is mislabelled.

We have corrected the legend (now Figure S3D).

**Reviewer #3 (Recommendations For The Authors):**
Protein homeostasis and lipid homeostasis are both are important for maintaining cellular functions. However, the crosstalk remains largely unknown. The manuscript entitled as "Impairment of lipid homoeostasis causes accumulation of protein aggregates in the lysosome" deals with this interesting topic. An important link between lysosomal protein aggregation and sphingolipids/cholesterol esters metabolism were discovered. The topic belonging to the Cell Biology domain also falls into the aims and scope of eLife. Here are the revisions I recommend:(1) From lipidomics analysis, a remarkable correlation between levels of sphingomyelin and cholesterol ester and ProteoStat staining was found. Could the authors explain how sphingomyelin and cholesterol ester are quantified? The two lipids are not included as internal standards from the lipidomics experiment.

Sphingomyelin and cholesterol ester internal standards are included in the Avanti 330707 SPLASH LIPIDOMIX Mass Spec Standard, which was supplied at 3% v/v to the MeOH/H2O cell lysis buffer. We have amended the Methods section to clarify this.

(2) Could the authors perhaps delete Figure 1B and show it on Figure 2A only? There is no need to show the same figure two times. The threshold of both False Discovery Rate and Median Enrichment needs to be added. From Figure 2A, the Lysosomal hydrolases (GBA, LIPA, GALC) seems located in statistically insignificant region. Based on previous studies, the GBA could have an effect on sphingolipid levels, then how to explain that sphingomyelin was highly correlated with ProteoSate staining?

We have combined the two volcano plots into a single figure (now Figure 1D), and added a line to help visualize the gene effects while considering the combined contribution of FDR and enrichment. Individual lysosomal hydrolases indeed have insignificant effects on ProteoStat and this is discussed in the main text as having relatively constrained impacts on the general lipidome. For example, while GBA and GALC KDs can lead to accumulation of their immediate substrates (glucosylceramide and galactosylceramide, respectively), they do not directly impinge on sphingomyelin.

(3) The authors show the corelation between ProteoState staining and different lipids/lipid classes in Figure 3B and Figure S3A. It is not necessary to show the corelation with individual lipids (such as sphingomyelin(d18:1/24:0) and cholesterol ester(18:2)). The corelation with full collection of lipid classes would be more representative, which is only list in Figure 3B and Figure S3A. It is suggested to add the information of how many individual lipids in each chass are used for the correlation analysis. Replace Figure 3A to Figure S3A, and put Figure 3A as supplementary figure are suggested.

We decided to retain the correlation of two individual lipids (a sphingomyelin and a cholesterol ester species) with ProteoStat as examples to illustrate with clarity how we obtained the class-wide comparison. The number of individual lipids included in each class for correlation analysis is now included in Figures 2F and S3A.

(4) The authors state that lipid uptake and metabolism modulate proteostasis. However, only cholesterol and LDL were tested. It would be more precise to state as cholesterol uptake and metabolism modulate proteostasis. In addition, sphingolipids and cholesterol esters accumulate with increased lysosomal protein aggregation. It would be interesting to see the effects of sphingolipids uptake, since sphingolipids are correlated with proteostasis better than cholesterol.

We attempted to add back specific sphingolipids to assess sufficiency. However, we found it challenging to ensure that these lipids were distributed to the correct subcellular locations at physiologically relevant levels. Without this crucial information, it was difficult to draw any conclusions about the sufficiency of the sphingolipids we tested to impair proteostasis.

Alonso A, Goñi FM. 2018. The Physical Properties of Ceramides in Membranes. Annu Rev Biophys 47:633–654. doi:10.1146/annurev-biophys-070317-033309

Ikonen E, Zhou X. 2021. Cholesterol transport between cellular membranes: A balancing act between interconnected lipid fluxes. Dev Cell 56:1430–1436. doi:10.1016/j.devcel.2021.04.025

Niekamp P, Scharte F, Sokoya T, Vittadello L, Kim Y, Deng Y, Südhoff E, Hilderink A, Imlau M, Clarke CJ, Hensel M, Burd CG, Holthuis JCM. 2022. Ca2+-activated sphingomyelin scrambling and turnover mediate ESCRT-independent lysosomal repair. Nat Commun 13:1875. doi:10.1038/s41467-022-29481-4